# Spatiotemporal constraints on optogenetic inactivation in cortical circuits

Nuo Li[1,2]*, Susu Chen[2], Zengcai V Guo[2,3], Han Chen[3], Yan Huo[3], Hidehiko K Inagaki[2], Guang Chen[1], Courtney Davis[1,2], David Hansel[4], Caiying Guo[2], Karel Svoboda[2]*

[1]Department of Neuroscience, Baylor College of Medicine, Houston, United States; [2]Janelia Research Campus, Ashburn, United States; [3]School of Medicine, Tsinghua University, Beijing, China; [4]Center of Neurophysics, Physiology and Pathologies, CNRS-UMR8119, Paris, France

**Abstract** Optogenetics allows manipulations of genetically and spatially defined neuronal populations with excellent temporal control. However, neurons are coupled with other neurons over multiple length scales, and the effects of localized manipulations thus spread beyond the targeted neurons. We benchmarked several optogenetic methods to inactivate small regions of neocortex. Optogenetic excitation of GABAergic neurons produced more effective inactivation than light-gated ion pumps. Transgenic mice expressing the light-dependent chloride channel GtACR1 produced the most potent inactivation. Generally, inactivation spread substantially beyond the photostimulation light, caused by strong coupling between cortical neurons. Over some range of light intensity, optogenetic excitation of inhibitory neurons reduced activity in these neurons, together with pyramidal neurons, a signature of inhibition-stabilized neural networks ('paradoxical effect'). The offset of optogenetic inactivation was followed by rebound excitation in a light dose-dependent manner, limiting temporal resolution. Our data offer guidance for the design of in vivo optogenetics experiments.

*For correspondence:
nuol@bcm.edu (NL);
svobodak@janelia.hhmi.org (KS)

## Introduction

The cerebral cortex consists of dozens of distinct areas (*Dong, 2008*; *Felleman and Van Essen, 1991*; *Paxinos and Watson, 1997*). Each brain region in turn contains multiple cell types (*Rudy et al., 2011*; *Tasic et al., 2018*; *Zeng and Sanes, 2017*). An important goal in neuroscience is to link dynamics in neural circuits to neural computation and behavior. Much of what we know about localization of cortical function comes from loss-of-function studies. Classically, lesions (*Lashley, 1931*; *Mishkin and Ungerleider, 1982*; *Newsome and Wurtz, 1988*), pharmacological inactivation (*Guo et al., 2017*; *Hikosaka and Wurtz, 1985*; *Krupa et al., 1999*), or cooling (*Long and Fee, 2008*; *Ponce et al., 2008*) have been used to silence activity in small (>1 mm$^3$) regions of tissue. The development of optogenetics has allowed rapid and reversible silencing of neuronal activity (*Deisseroth, 2015*). Concurrently, there has been significant progress in creating genetic access to specific neuronal populations (*Luo et al., 2018*). Various transgenic Cre driver mouse lines target subtypes of cortical neurons (*Gerfen et al., 2013*; *Gong et al., 2007*; *Harris et al., 2014*; *Taniguchi et al., 2011*). Together with Cre-dependent reporter lines that endogenously express optogenetic effector proteins (*Madisen et al., 2015*; *Madisen et al., 2012*; *Zhao et al., 2011*), and viral delivery methods (*Atasoy et al., 2008*; *Chatterjee et al., 2018*; *Deverman et al., 2016*; *Dimidschstein et al., 2016*; *Luo et al., 2018*; *Tervo et al., 2016*; *Wickersham et al., 2007*), these technologies have advanced the precision with which manipulation experiments can be carried out.

Optogenetic loss-of-function experiments rely on two schemes (*Wiegert et al., 2017*). '*Direct photoinhibition*' involves light-gated Cl-/H+ pumps that hyperpolarize neurons (*Brown et al., 2018*;

*Chow et al., 2010*; *Chuong et al., 2014*; *Han and Boyden, 2007*; *Zhang et al., 2007*) and light-gated Cl- channels that produce hyperpolarization and shunting inhibition (*Berndt et al., 2014*; *Govorunova et al., 2015*; *Wietek et al., 2014*). Direct photoinhibition silences genetically-defined neuronal populations. A second scheme is '*ChR-assisted photoinhibition*', which relies on activation of excitatory channelrhodopsins (ChR) expressed in GABAergic neurons. Photostimulation of the GABAergic neurons potently inhibits local pyramidal neurons and thereby removes excitatory output from the photoinhibited brain region (*Cardin et al., 2009*; *Guo et al., 2014b*; *Olsen et al., 2012*). ChR-assisted photoinhibition has been applied in transgenic mice (such as VGAT-ChR2-EYFP) that express ChR2 in GABAergic neurons (*Zhao et al., 2011*); this does not require complex crosses and Cre-mediated expression can be used for other genetic labels. Alternatively, specific variants of ChR (e.g. ChR2 vs. red-shifted variants; *Hooks et al., 2015*; *Klapoetke et al., 2014*; *Lin et al., 2013*) can be targeted to specific GABAergic neurons (e.g. PV vs. Sst neurons) using interneuron-specific Cre lines.

Both direct photoinhibition and ChR-assisted photoinhibition have been widely used in mice to reveal the involvement of brain areas and neuronal populations in specific phases of behavior (*Goard et al., 2016*; *Guo et al., 2015*; *Guo et al., 2014b*; *Hanks et al., 2015*; *Kwon et al., 2016*; *Li et al., 2015*; *Li et al., 2016*; *Mathis et al., 2017*; *Morandell and Huber, 2017*; *Resulaj et al., 2018*; *Sachidhanandam et al., 2013*). However, neural circuits have local and long-range connections (*Harris and Shepherd, 2015*; *Hooks et al., 2011*; *Hooks et al., 2013*; *Kato et al., 2017*; *Lefort et al., 2009*; *Mao et al., 2011*; *Ozeki et al., 2009*; *Xue et al., 2014*). Strong and nonlinear coupling between neurons, in addition to the photostimulus itself, can affect the spatial and temporal patterns in changes of activity. As a result, unexpected effects of optogenetic manipulations are common. For example, activating and inhibiting Sst+ and PV+ neurons can produce complex and asymmetric effects on excitatory neuron stimulus selectivity in visual cortex (*Adesnik et al., 2012*; *Atallah et al., 2012*; *Lee et al., 2012*; *Wilson et al., 2012*) and auditory cortex (*Phillips and Hasenstaub, 2016*; *Seybold et al., 2015*). The effect depends on brain state (*Phillips and Hasenstaub, 2016*), nonlinearity (*Phillips and Hasenstaub, 2016*), connectivity of the neural circuits (*Seybold et al., 2015*), or strength and duration of the light manipulation (*Atallah et al., 2014*; *El-Boustani et al., 2014*; *Lee et al., 2014*). Theoretical studies of networks of excitatory and inhibitory neurons predict counterintuitive responses of neural circuits to perturbations. For example, excitation of cortical interneurons can cause a decrease in activity of the same interneurons ('paradoxical effect'), which arises from the interplay between recurrent excitation and inhibition (*Kato et al., 2017*; *Sanzeni, 2019*; *Litwin-Kumar et al., 2016*; *Ozeki et al., 2009*; *Pehlevan and Sompolinsky, 2014*; *Rubin et al., 2015*; *Sadeh et al., 2017*; *Tsodyks et al., 1997*).

Optogenetic manipulations of a brain region can also impact activity of downstream regions in complex ways. Activity in a brain region can be robust to perturbation of one strongly connected brain region. For example, inactivating one side of anterior lateral motor cortex has little effect on activity in the contralateral hemisphere, despite abundant interhemispheric connections (*Li et al., 2016*). Perturbation of other connected brain regions can have dramatic effects. For example, inactivating thalamus causes a rapid and complete collapse of cortical activity (*Guo et al., 2017*; *Reinhold et al., 2015*). Perturbation experiments using behavior as readout must be interpreted in terms of measured changes in neuronal activity throughout the circuit. The direct and indirect effects of circuit manipulations on neuronal activity are rarely measured, particularly in vivo.

We characterized several optogenetic methods to locally silence somatosensory and motor cortex, either using direct photoinhibition or ChR-assisted photoinhibition. In addition, we measured the wavelength-dependent spread of light and its effects on the spatial spread of inactivation. Many light-gated opsins (such as ChR2) are excited by blue light. But blue light is highly scattered and absorbed by blood (*Chow et al., 2010*; *Stujenske et al., 2015*; *Wiegert et al., 2017*; *Yizhar et al., 2011*; *Yona et al., 2016*). Red light is less subject to hemoglobin absorption (*Svoboda and Block, 1994*), and thus can propagate further and produces less local heating (*Liu et al., 2015*; *Stujenske et al., 2015*; *Wiegert et al., 2017*). Red-shifted opsins (e.g. ReaChR [*Lin et al., 2013*], Chrimson [*Klapoetke et al., 2014*], Jaws [*Chuong et al., 2014*]) could enable noninvasive manipulations of deep brain regions. However, direct measurements of light propagation in the intact brain is scarce (*Guo et al., 2014b*; *Ranganathan et al., 2018*; *Yizhar et al., 2011*; *Yona et al., 2016*), and measurements of the distribution of light intensity in vivo are missing. We directly measured the

spatial profile of light intensity in cortex at two commonly used wavelengths (blue and orange, 473 and 594 nm) using a photobleaching assay (*Guo et al., 2014b*).

We found that the spatial resolution of optogenetic silencing is approximately one millimeter, with silencing spreading further than the photostimulation light. The length scale of photoinhibition is caused by strong coupling within cortical circuits: loss of activity in a focal region withdraws input to other layers and surrounding regions, resulting in loss of activity in both excitatory and inhibitory populations in the surround, as predicted by network models (*Litwin-Kumar et al., 2016*; *Rubin et al., 2015*; *Sadeh et al., 2017*; *Tsodyks et al., 1997*). Offset of the photostimulus was typically followed by rebound excitation, which limits the temporal resolution of photoinhibition. Our data outline spatial and temporal constraints of optogenetics manipulations in vivo and provide guidance to the design of loss-of-function experiments.

## Results

### Optogenetic inactivation

We examined eight different optogenetic methods to inactivate cortical activity in awake mice (*Table 1*, *Figure 1A*).

For ChR-assisted photoinhibition we photostimulated excitatory opsins in GABAergic interneurons to drive inhibition in nearby pyramidal neurons. We used transgenic mice that expressed ChR2 in GABAergic neurons (VGAT-ChR2-EYFP), or in parvalbumin-positive (PV) interneurons (PV-IRES-Cre X Ai32), or in somatostatin-positive (Sst) interneurons (*Sst*-IRES-Cre X Ai32). In addition, we photostimulated a red-shifted channelrhodopsin (ReaChR) in PV neurons (PV-IRES-Cre X ReaChR) (*Hooks et al., 2015*; *Lin et al., 2013*). We also induced photoinhibition with a Cre-dependent ChR2 virus in PV-IRES-Cre mice. For direct photoinhibition we photostimulated inhibitory opsins in pyramidal neurons, including the ion pumps Archaerhodopsin (Arch, *Emx1*-IRES-Cre X Ai35D) and Jaws (*Emx1*-IRES-Cre X Camk2a-tTA X Ai79) (*Chow et al., 2010*; *Chuong et al., 2014*; *Madisen et al., 2015*).

**Table 1.** A list of photoinhibition methods tested in this study.

| Methods | Mouse (JAX #) | Reagents | Wavelength | Brain region |
|---|---|---|---|---|
| *ChR-assisted photoinhibition* | | | | |
| ChR2 in all GABAergic neurons | VGAT-ChR2-EYFP or *Slc32a1-COP4\*H134R/EYFP* (014548) | | 473 nm | Somatosensory cortex, ALM |
| ChR2 in PV expressing neurons | PV-IRES-Cre or *Pvalb*-IRES-Cre (008069) x Ai32 (012569) | | 473 nm | Somatosensory cortex |
| ChR2 in Sst neurons | *Sst*-IRES-Cre (013044) x Ai32 (012569) | | 473 nm | Somatosensory cortex |
| ReaChR in PV neurons | PV-IRES-Cre (008069) x R26-CAG-LSL-ReaChR-mCit (026294) | | 594 nm | ALM |
| ChR2 virally delivered to local PV neurons | PV-IRES-Cre (008069) | AAV2/1-CAG-FLEX-ChR2-tdTomato-WPRE (UPenn Viral Core, AV-1-ALL864) | 473 nm | Somatosensory cortex |
| *Direct photoinhibition* | | | | |
| Arch in excitatory neurons | *Emx1*-IRES-Cre (005628) x Ai35D (012735) | | 594 nm | Somatosensory cortex |
| Jaws in excitatory neurons | *Emx1*-IRES-Cre (005628) x Ai79D (023529) x Camk2-tTA (003010) | | 594 nm | Somatosensory cortex |
| GtACR1 (somatic targeting) (*Mahn et al., 2018*) in excitatory neurons | *Emx1*-IRES-Cre (005628) x R26-CAG-LNL-GtACR1-ts-FRed-Kv2.1 (033089) | | 473 nm, 635 nm | Somatosensory cortex, primary motor cortex, ALM |

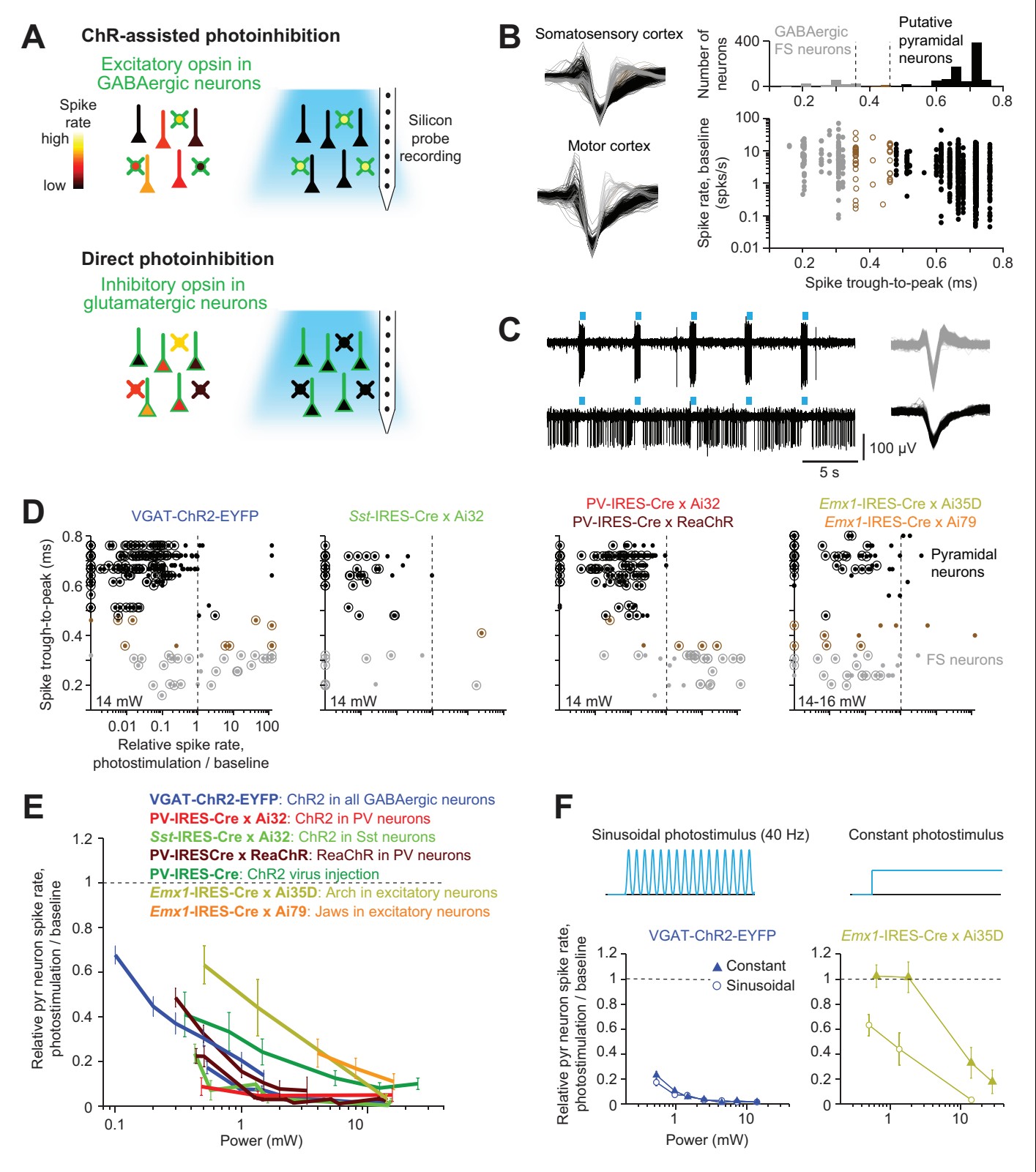

**Figure 1.** Optogenetic inactivation and cell-type specific recording. (**A**) Summary of inactivation methods. ChR-assisted photoinhibition was induced by photostimulating excitatory opsins in various GABAergic interneurons. Direct photoinhibition was achieved by photostimulating inhibitory opsins in pyramidal neurons. (**B**) Cell-type classification based on spike waveform. *Left,* spike waveforms for putative FS neurons (gray) and putative pyramidal neurons (black) in two different cortical areas. *Right,* histogram of trough-to-peak spike durations (top) and relationship to baseline spike rate (bottom).
*Figure 1 continued on next page*

Figure 1 continued

Neurons were classified into putative GABAergic fast spiking (FS) neurons and pyramidal neurons based on spike width (Materials and methods). Neurons that could not be classified (brown) were excluded from analysis. (C) Silicon probe recordings during photostimulation in a VGAT-ChR2-EYFP mouse. *Top*, a putative FS neuron. *Bottom*, a putative pyramidal neuron. *Right*, corresponding spike waveforms. (D) Effects of photostimulation on cell types defined by spike waveform. Dots correspond to individual neurons. Spike rates of each neuron during photostimulation were normalized to its baseline ('relative spike rate', Materials and methods). Neurons with significant spike rate changes ($p<0.05$, two-tailed *t*-test) are highlighted by circles. Gray, putative FS neurons; black, putative pyramidal neurons; brown, neurons that could not be classified. (E) Relative spike rate as a function of laser power for different photoinhibition methods. Pyramidal neurons within 0.4 mm from laser center across all cortical depth. Mean ± s.e.m. across neurons, bootstrap (Materials and methods). Two datasets (lines) are shown for VGAT-ChR2-EYFP: the line spanning 0.5–10.5 mW shows data from barrel cortex, n = 153; the line spanning 0.1–1.5 mW shows data from ALM, n = 188; PV-IRES-Cre x Ai32, data from barrel cortex, n = 16; *Sst*-IRES-Cre x Ai32, data from barrel cortex, n = 65; two datasets (lines) are shown for PV-IRES-Cre x ReaChR: the line spanning 0.5–10.5 mW shows data from barrel cortex, n = 211; the line spanning 0.3–3 mW shows data from ALM, n = 55; PV-IRES-Cre, ChR2 virus injection, data from barrel cortex, n = 78; *Emx1*-IRES-Cre x Ai35D, data from barrel cortex, n = 26; *Emx1*-IRES-Cre x Camk2a-tTA x Ai79, data from barrel cortex, n = 176. (F) Effect of photostimulus profile on photoinhibition. *Top*, 40 Hz sinusoid photostimulus and constant photostimulus. *Bottom*, relative spike rate as a function of laser power for ChR-assisted photoinhibition (left) and direct photoinhibition mediated by Arch (right). Pyramidal neurons within 0.4 mm from laser center across all cortical depth. Mean ± s.e.m. across neurons, bootstrap (Materials and methods). 40 Hz sinusoid photostimulus is the standard photostimulus used in this study.

We measured neural activity in the vicinity of the photostimulus using silicon probe recordings. Measurements were performed in somatosensory cortex and motor cortex; these two cortical regions are examples of sensory and frontal cortex, which differ in laminar connectivity (*Hooks et al., 2011*) and in their connections with thalamus (*Hooks et al., 2013*) (Materials and methods). In both brain areas the distribution of single-unit spike width was bimodal (*Figure 1B*). Units with narrow spikes were putative fast spiking (FS) neurons and likely expressed parvalbumin (*Cardin et al., 2009*; *Guo et al., 2014b*; *Olsen et al., 2012*; *Resulaj et al., 2018*). Neurons with wide spikes were likely mostly glutamatergic pyramidal neurons, and we refer to this population as putative pyramidal neurons.

These classifications were consistent with spike rate changes observed during photostimulation (*Figure 1C–D*). In mice expressing excitatory opsins in FS GABAergic neurons (VGAT-ChR2-EYFP, PV-IRES-Cre X Ai32, and PV-IRES-Cre X ReaChR mice), FS neurons were activated by photostimulation, whereas putative pyramidal neurons were inhibited (*Figure 1D*). In VGAT-ChR2-EYFP mice a subset of FS neurons were inhibited rather than excited, likely caused by activation of other GABAergic neurons that were photostimulated in these mice. In *Sst*-IRES-Cre X Ai32 mice, both FS neurons and putative pyramidal neurons were inhibited, consistent with Sst neurons inhibiting both PV and pyramidal neurons (*Lee et al., 2013*; *Pfeffer et al., 2013*). Sst neurons were rare in extracellular recordings, with only two neurons showing clear photostimulus-induced increases in spike rate. Putative Sst neurons could not be unambiguously separated from either FS neurons or putative pyramidal neurons based on their spike waveforms (*Figure 1D*) (*Yu et al., 2019*). In *Emx1*-IRES-Cre X Ai35D and *Emx1*-IRES-Cre X Camk2a-tTA X Ai79 mice, optogenetic hyperpolarization of pyramidal neurons reduced activity of putative pyramidal neurons and FS neurons. The inhibition of FS neurons was likely caused by withdrawal of excitatory input from pyramidal neurons.

We measured the light sensitivity of different optogenetic inactivation methods by computing the 'relative spike rate': the spike rate of putative pyramidal neurons during inactivation, averaged across neurons, divided by the baseline spike rate (Materials and methods). Near the center of the photostimulus (<0.4 mm from laser center), ChR-assisted photoinhibition in transgenic mice produced strong inactivation, with >80% of reduction in spike rate for putative pyramidal neurons with low laser powers (0.5–1.5 mW, *Figure 1E*). The powers used here are far below the levels that produce heating in tissue and non-specific effects on neural activity (*Christie et al., 2012*; *Owen et al., 2019*; *Stujenske et al., 2015*). ChR2 virus mediated photoinhibition in PV-IRES-Cre mice produced slightly weaker silencing. Arch and Jaws-mediated photoinhibition required roughly 10-fold higher laser power to achieve similar silencing compared to ChR-assisted photoinhibition (*Figure 1E*). For Arch-mediated photoinhibition, temporally-modulated photostimuli were more effective in silencing activity than constant photostimuli at the same average power (*Figure 1F*).

We also photoactivated the *Guillardia theta* anion channel rhodopsin 1 (GtACR1) to hyperpolarize and shunt the membranes of pyramidal neurons (*Govorunova et al., 2015*). We generated a Cre reporter mouse, expressing a soma localized GtACR1 (*Mahn et al., 2018*) driven by the CAG

promoter targeted to the Rosa26 locus (*Hooks et al., 2015*; *Madisen et al., 2010*; *Muzumdar et al., 2007*) (*Figure 2A*) (R26-CAG-LNL-GtACR1-ts-FRed-Kv2.1, JAX #033089). We expressed soma localized GtACR1 in cortical excitatory neurons by crossing the reporter mouse to *Emx1*-IRES-Cre. GtACR1 expression was targeted to the soma, but weak expression was still seen in axons (*Figure 2A and B*). Under blue light (473 nm) illumination, GtACR1 produced the most potent photoinhibition near the laser center, with >80% of the spikes silenced at 0.1 mW (*Figure 2C*). Previous studies have shown that GtACR1 and other Cl- channels induce spiking in axons, caused by a positively shifted chloride reversal potential in the axon (*Mahn et al., 2018*; *Messier et al., 2018*). Below 0.2 mW, photostimulation of soma localized GtACR1 induced little excitation (*Mahn et al., 2018*; *Messier et al., 2018*) (*Figure 2D*). At higher laser powers, transient short-latency excitation was apparent, which was likely due to axonal excitation. GtACR1 mediated photoinhibition could also be induced using red light (*Figures 2C*, 635 nm) (albeit at higher light intensities), which could

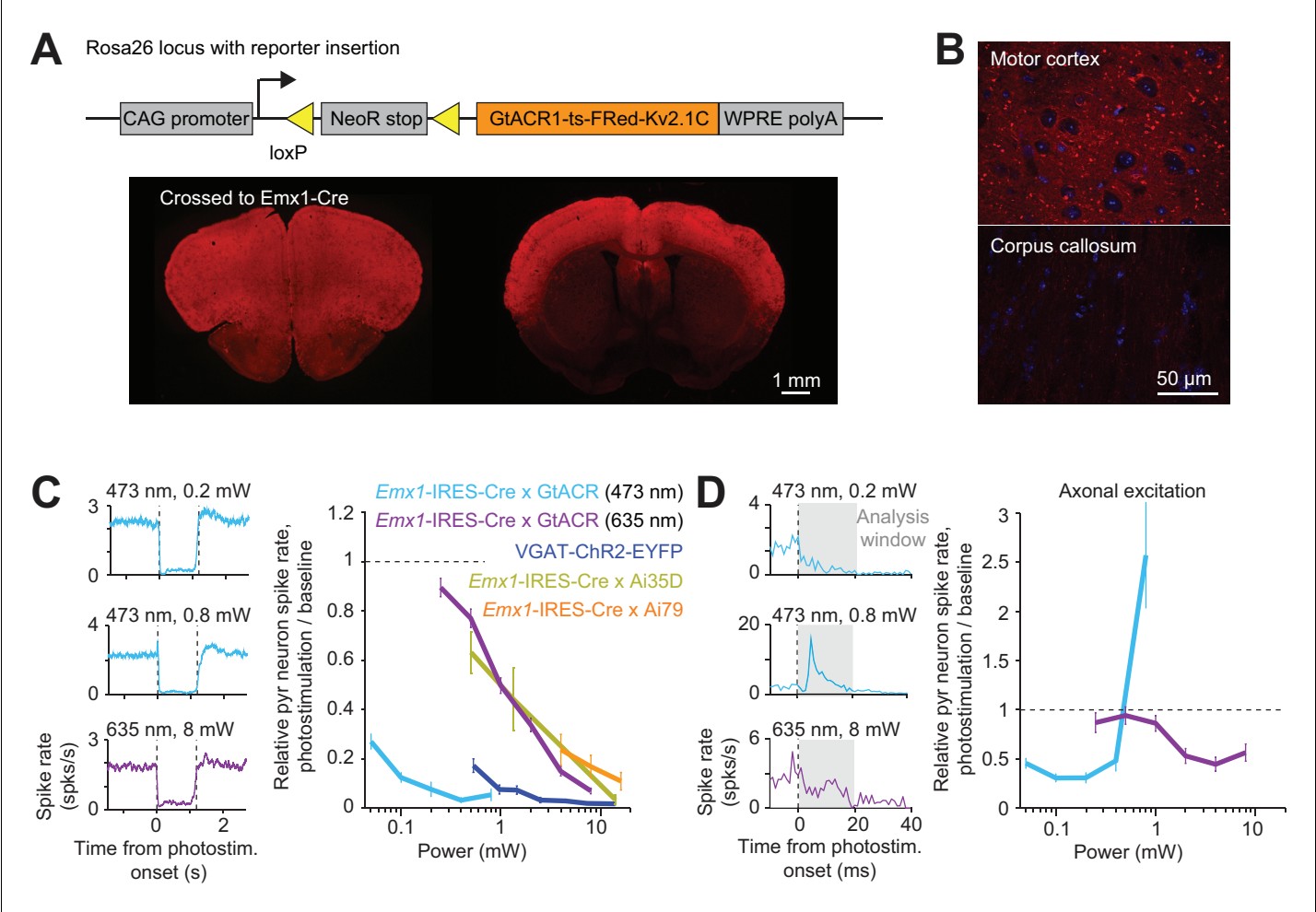

**Figure 2.** Direct photoinhibition with a GtACR reporter mouse. (A) *Top*, generation of the Cre-dependent GtACR reporter line. Construct includes loxP sites, and GtACR1-ts-FRed-Kv2.1C with WPRE, driven by CAG promoter targeted to the Rosa26 locus. *Bottom*, cross to *Emx1*-IRES-Cre mice. GtACR1 is expressed in cortical excitatory neurons. (B) Confocal images showing dense expression of GtACR1 in cortex and low levels of expression in the corpus callosum, implying low trafficking of GtACR1 to axons. (C) *Left*, mean peristimulus time histogram (PSTH, 50 ms bin) for pyramidal neurons with blue and red photostimuli. Dashed lines, photostimulus onset and offset. Pyramidal neurons within 0.4 mm from laser center across all cortical depth. Blue light photostimulation, n = 335 neurons from ALM; Red light photostimulation, n = 285 neurons from ALM. *Right*, relative spike rate as a function of laser power. Mean ± s.e.m. across neurons, bootstrap (Materials and methods). VGAT-ChR2-EYFP, *Emx1*-IRES-Cre x Ai35D, *Emx1*-IRES-Cre x Camk2a-tTA x Ai79, data from *Figure 1E* are replotted here for reference. (D) *Left*, mean PSTH (1 ms bin) at the onset of the photostimulation. Same data as in (C). Axonal excitation was induced at 0.8 mW laser power for blue light photostimulation. Right, relative spike rate during the first 20 ms of photostimulation onset.

facilitate photoinhibition of deep brain regions. Expression of GtACR1 appears to be the most sensitive method for inactivation, although attention has to be paid to avoid axonal excitation.

## Spatial profile of light intensity

To characterize the spread of inactivation we first measured the spatial profile of the photostimulus, that is, the light intensity in the tissue. At the surface of the brain the photostimulus was a stationary collimated laser beam with an approximately Gaussian profile (diameter at $4\sigma$, 400 µm) (Materials and methods). In the brain, light is scattered and absorbed, primarily by blood. As a substitute for light intensity we measured the three-dimensional profile of photobleaching of fluorescent proteins in transgenic mice. Because neurons are large (100's of micrometers) and fluorescent proteins diffuse rapidly in the cytoplasm (*Swaminathan et al., 1997*), we used fluorescent proteins targeted to neuronal nuclei. We measured the spatial distribution of fluorescence after prolonged light exposure, which caused pronounced photobleaching at the center of the photostimulus. For blue light (473 nm), we used transgenic mice expressing GFP in the nuclei of cortical excitatory neurons (Rosa-LSL-H2B-GFP crossed to *Emx1*-IRES-Cre) (*He et al., 2012*). For orange light (594 nm), we used transgenic mice expressing mCherry (Rosa-LSL-H2B-mCherry crossed to *Emx1*-IRES-Cre) (*Peron et al., 2015*).

Blue light induced photobleaching in a confined region near the photostimulus (*Figure 3A*). Photobleaching increased with the light dose (*Figure 3B*). We measured photobleaching by averaging the fluorescence change ($\Delta F/F_0$) within a small region near the photostimulus center relative to the baseline fluorescence in regions far away from the laser center ($F_0$, *Figure 3C*). At our illumination conditions, photobleaching is a linear process. The rate of photobleaching is therefore proportional to light intensity and the relationship between $\Delta F/F_0$ and the light dose (light intensity x time) is exponential (*Figure 3C*). Calibration experiments allowed us to infer the spatial profile of light intensity in tissue, which is independent of light dose (*Figure 3D*, Materials and methods). Light intensity attenuated rapidly as a function of depth, to less than 20% at 500 µm below the surface of the brain (*Figure 3D*; depth at half max, 300 µm). Laterally, light intensity dropped off to 50% at 135 µm (*Figure 3D*). These data show that blue light is confined in cortical tissue, consistent with simulations (*Stujenske et al., 2015*).

In contrast to blue light, orange light propagated much further (depth at half max, 846 µm), penetrating all layers of cortex and illuminating a cubic millimeter of volume (*Figure 3E–G*), 28 times larger than for blue light. Because orange light (594 nm) is still absorbed by hemoglobin (*Svoboda and Block, 1994*), longer wavelength light (e.g. 630 nm) will likely penetrate still deeper into tissue than orange light.

We note that the photobleaching experiments are relatively noisy and have limited dynamic range (*Figure 3B,F*). As a result, our measurement likely misses a long tail in light intensity that is still able to activate light gated channels. For example, at high powers FS interneurons can be activated up to 1.5 mm from the laser center, even with blue light (see below and Figure 5B,C,D).

## Spatial profile of optogenetic inactivation

We measured photoinhibition across cortical layers when photostimulating GABAergic neurons using blue or orange light (*Figure 4A–B*). Recordings were made from the whisker representation area of the somatosensory cortex (barrel cortex) near the center of the photostimulus. Photostimulation of ChR2 in GABAergic neurons produced nearly uniform inhibition of pyramidal neurons across cortical layers (*Figure 4A*, VGAT-ChR2-EYFP, *Sst*-IRES-Cre X Ai32, and PV-IRES-Cre X Ai32), despite limited penetration of blue light in tissue (*Figure 3*). Photoinhibition removed the majority of spikes (>80%) in a column across a wide range of laser powers (1.5–14 mW), indicating that photoinhibition is an effective method for local inactivation.

A notable exception was seen at low laser powers (0.5 mW) in *Sst*-IRES-Cre X Ai32 mice. Selective excitation of Sst neurons produced a disinhibition of excitatory neurons around layer 4 (*Figure 4A*, , *Sst*-IRES-Cre X Ai32). This disinhibition in layer four was likely mediated by Sst neurons that inhibit PV neurons. Activation of Sst neurons therefore removed a major source of inhibition onto excitatory neurons (*Xu et al., 2013*). Thus, driving inhibition can effectively drive excitatory neurons (*Pfeffer et al., 2013*; *Yu et al., 2019*) highlighting the complexities of optogenetic manipulations in actual neural networks.

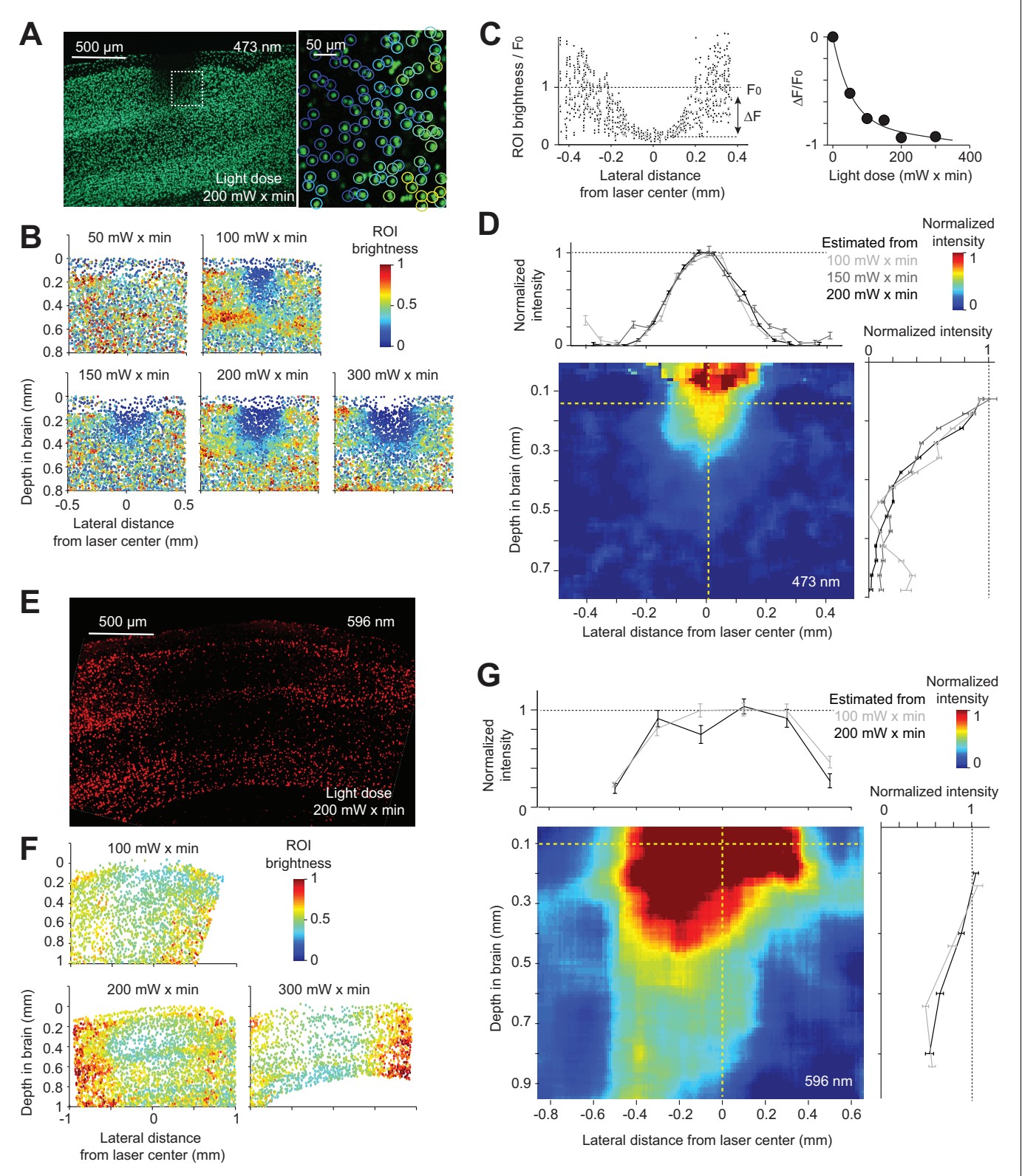

**Figure 3.** Spatial profile of light in the cortex. (**A**) Photobleaching was induced in a transgenic mouse line expressing GFP in the nuclei of excitatory neurons (Rosa-LSL-H2B-GFP crossed to *Emx1*-IRES-Cre; Materials and methods). *Left*, coronal section showing an example photobleaching site. Light dose, 200 mW x min. The photostimulus was a laser beam with a Gaussian profile (diameter at 4σ, 0.4 mm) at the brain surface. *Right*, a confocal image showing a region highlighted by the dash line. GFP intensity was measured in regions of interest (ROI) around individual nuclei (circles). Color indicates
*Figure 3 continued on next page*

*Figure 3 continued*

ROI fluorescence intensity. (**B**) Photobleaching assay measuring the spread of blue light in the neocortex. Photobleaching was induced with different light doses (as indicated). Dots, individual nuclei. Color indicates ROI fluorescence intensity. (**C**) *Left*, ROI fluorescence intensity as a function of lateral distance from the laser center (data from light dose 200 mW x min, at 0.2 mm depth). $F_0$, baseline ROI intensity, computed by averaging all ROIs far away from the laser center. $\Delta F$, ROI fluorescence intensity change caused by photobleaching, computed by averaging all ROIs near the laser center and subtracting $F_0$ (Materials and methods). *Right*, photobleaching ($\Delta F/ F_0$) at various light doses. Line, exponential fit to $\Delta F/ F_0$ as a function of light dose. (**D**) The estimated spatial profile of blue light in tissue. $\Delta F/ F_0$ was converted to light intensity using the exponential fit in (**C**) (see Material and methods). Light intensity is shown as a function of lateral distance (top) and cortical depth (right), along the yellow dashed lines. Mean ± s.e.m. across ROIs, bootstrap (Materials and methods). The color map shows the spatial profile estimated from the 200 mW x min light dose data. Estimates from other light doses produced similar spatial profiles. (**E**) - (**G**) Same as (**A**) - (**D**), but for orange light (594 nm) measured with photobleaching of mCherry (Material and methods).

Photostimulation of GABAergic neurons using orange light also induced uniform inhibition across cortical layers (*Figure 4B*, PV-IRES-Cre X ReaChR). Despite different propagation of blue and orange light in tissue (*Figure 3*), ChR-assisted photoinhibition was remarkably similar for both illumination wavelengths. At moderate laser power (1.5 mW), blue light excited FS neurons mainly in superficial layers (*Figure 4C*). Yet, loss of activity was present in both superficial and deep layers (see Discussion). Direct photoinhibition of excitatory neurons using GtACR1 or Jaws also inhibited activity across all layers (*Figure 4D–F*). Remarkably, the profile of silencing mediated by GtACR1 was also similar for blue and red light (*Figure 4D–E*), despite profound wavelength-dependent differences of light propagation in tissue (and much higher power levels required at the longer excitation wavelength).

We next characterized the lateral extent of photoinhibition by varying the location of the photostimulus relative to the recording site (*Figure 5A*). In VGAT-ChR2-EYFP mice, FS neurons near the laser center were activated in a dose-dependent manner (*Figure 5B*). The lateral spread of FS neuron excitation was dependent on laser power (at 0.5 mW, half-max width, 0.15 mm; at 14 mW, 1.5 mm, *Figure 5B*). These measurements show that a long tail in the light intensity profile can activate light gated channel more than 1 mm from the center of the laser, even for blue excitation light.

Photoinhibition of pyramidal neurons extended 1 mm beyond the activation profile of FS neurons (*Figure 5B*). For example, at 0.5 mW, pyramidal neuron activity was reduced even 1 mm away from the laser center where FS neurons were not activated. This spread of photoinhibition extends beyond typical sizes of dendritic and axonal arbors of GABAergic neurons (*Jiang et al., 2015*) and dendritic arbors of pyramidal neurons (*Oberlaender et al., 2012*; *Oswald et al., 2013*; *Shepherd et al., 2005*; *Yamashita et al., 2018*). In VGAT-ChR2-EYFP mice, ChR2 was expressed in all GABAergic neurons. We considered the possibility that the spread of photoinhibition was mediated by a specific subtype of GABAergic neuron. Among GABAergic interneurons, PV neurons have the most compact dendritic and axonal arbors (*Jiang et al., 2015*). We thus tested whether selectively photostimulating PV neurons would produce more spatially-restricted photoinhibition. However, a similarly broad photoinhibition profile was induced in PV-IRES-Cre X Ai32 mice (*Figure 5C*). We next sought to produce a broader photoinhibition by using orange light to illuminate a larger cortical volume in transgenic mice expressing ReaChR in PV neurons. However, the resulting photoinhibition profile was similar to those produced by photostimulating ChR2 in PV neurons using blue light (*Figure 5D*).

We examined other methods of optogenetic inactivation. First, photoinhibition by selective excitation of Sst neurons induced similarly broad photoinhibition (*Figure 5E*). Next, direct photoinhibition of pyramidal neurons using light-gated ion pumps (*Emx1*-IRES-Cre X Ai35D mice or *Emx1*-IRES-Cre X Camk2a-tTA X Ai79 mice) induced similarly broad photoinhibition (*Figure 5F and G*). ChR-assisted photoinhibition in a different brain region (anterior lateral motor cortex, ALM) also produced similarly broad spatial spread regardless of the method used to drive photoinhibition (*Figure 5H–I*, blue light photostimulation in VGAT-ChR2-EYFP mice, or orange light photostimulation in PV-IRES-Cre X ReaChR mice). Across all methods, the fractional reduction in spike rate near the laser center approximately predicted the lateral spread of photoinhibition (*Figure 5J*). These data (*Figures 4* and *5*) indicate that, for relatively localized photostimuli, the spatial resolution of photoinhibition laterally and axially is shaped by interactions between spatially extended activation of GABAergic interneurons with intrinsic properties of cortical circuits.

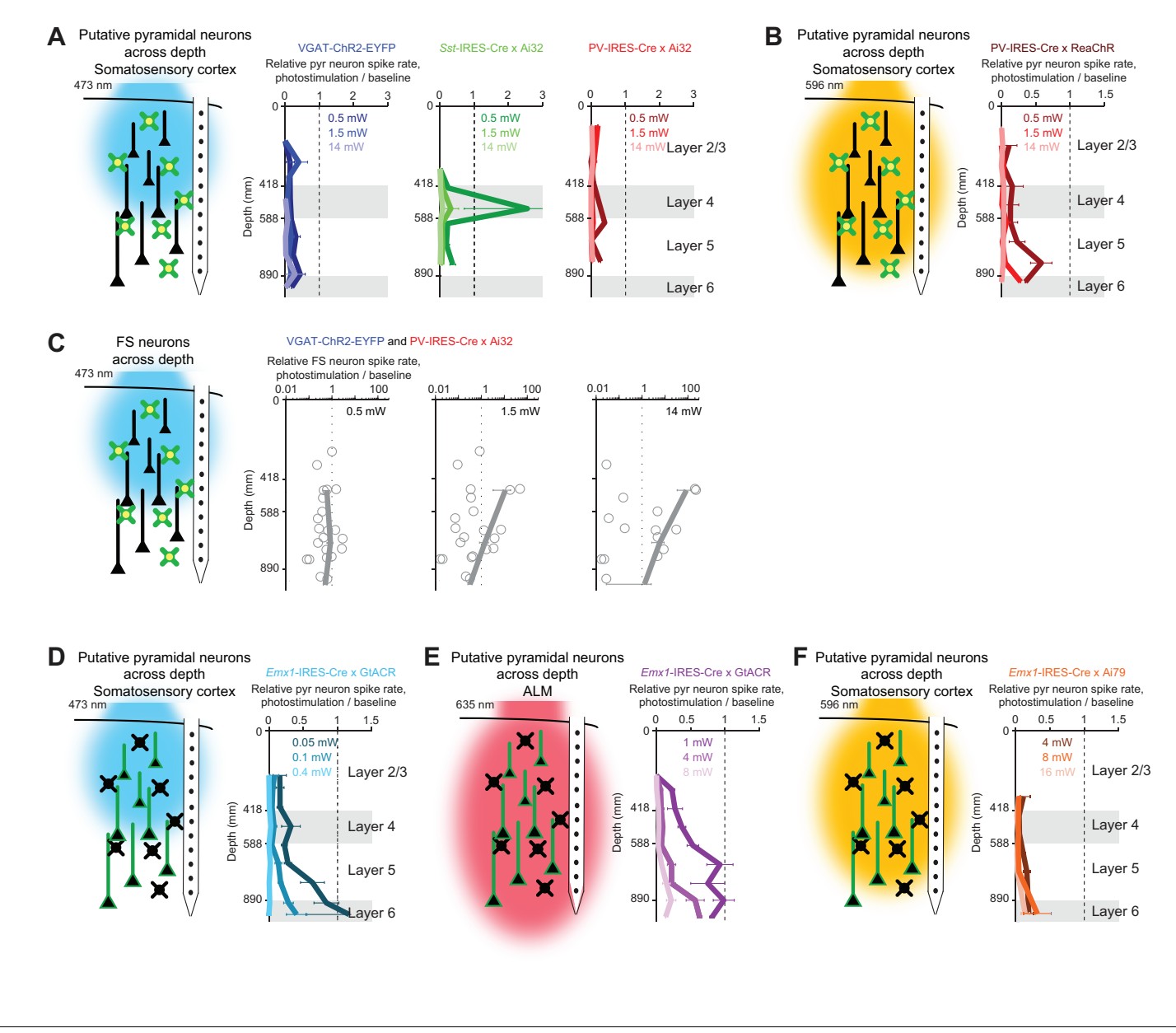

**Figure 4.** Photoinhibition across cortical layers. (**A**) *Left*, schematics, ChR-assisted photoinhibition using blue light. *Right*, relative spike rate across cortical depth for three laser powers (0.5, 1.5, 14 mW). Pyramidal neurons within 0.2 mm from laser center. Data from barrel cortex. Mean ± s.e.m. across neurons, bootstrap (Materials and methods). VGAT-ChR2-EYFP, n = 170; *Sst*-IRES-Cre x Ai32, n = 33; PV-IRES-Cre x Ai32, n = 61. (**B**) Same as (**A**), but for orange light photostimulation of red-shifted ChR in PV neurons (PV-IRES-Cre x R26-CAG-LSL-ReaChR-mCitrine, n = 95). (**C**) FS neuron responses during ChR-assisted photoinhibition using blue light. Relative spike rate for putative FS neurons across cortical depth for three laser powers (0.5, 1.5, 14 mW). Putative FS neurons within 1 mm from laser center. Data from barrel cortex and motor cortex are combined. Data from VGAT-ChR2-EYFP and PV-IRES-Cre x Ai32 mice are combined (n = 22). Circles, individual neurons; mean ± s.e.m. over neurons. (**D**) Same as (**A**), but for blue light photostimulation of light-gated Cl⁻ channel GtACR1 in pyramidal neurons (*Emx1*-IRES-Cre X GtACR, n = 198). (**E**) Same as (**A**), but for red light photostimulation of GtACR1 in pyramidal neurons (n = 236). (**F**) Same as (**A**), but for orange light photostimulation of light-gated Cl⁻ pump Jaws in pyramidal neurons (*Emx1*-IRES-Cre X Camk2a-tTA X Ai79, n = 176).

Higher spatial resolution (i.e. complete removal of spikes from smaller tissue volumes) is desirable for experiments investigating small brain regions. If the length scale of inactivation arises from intrinsic properties of cortical circuits, it would pose a fundamental constraint on the spatial resolution of neuronal inactivation. Our results suggest that silencing a small area also reduces activity in

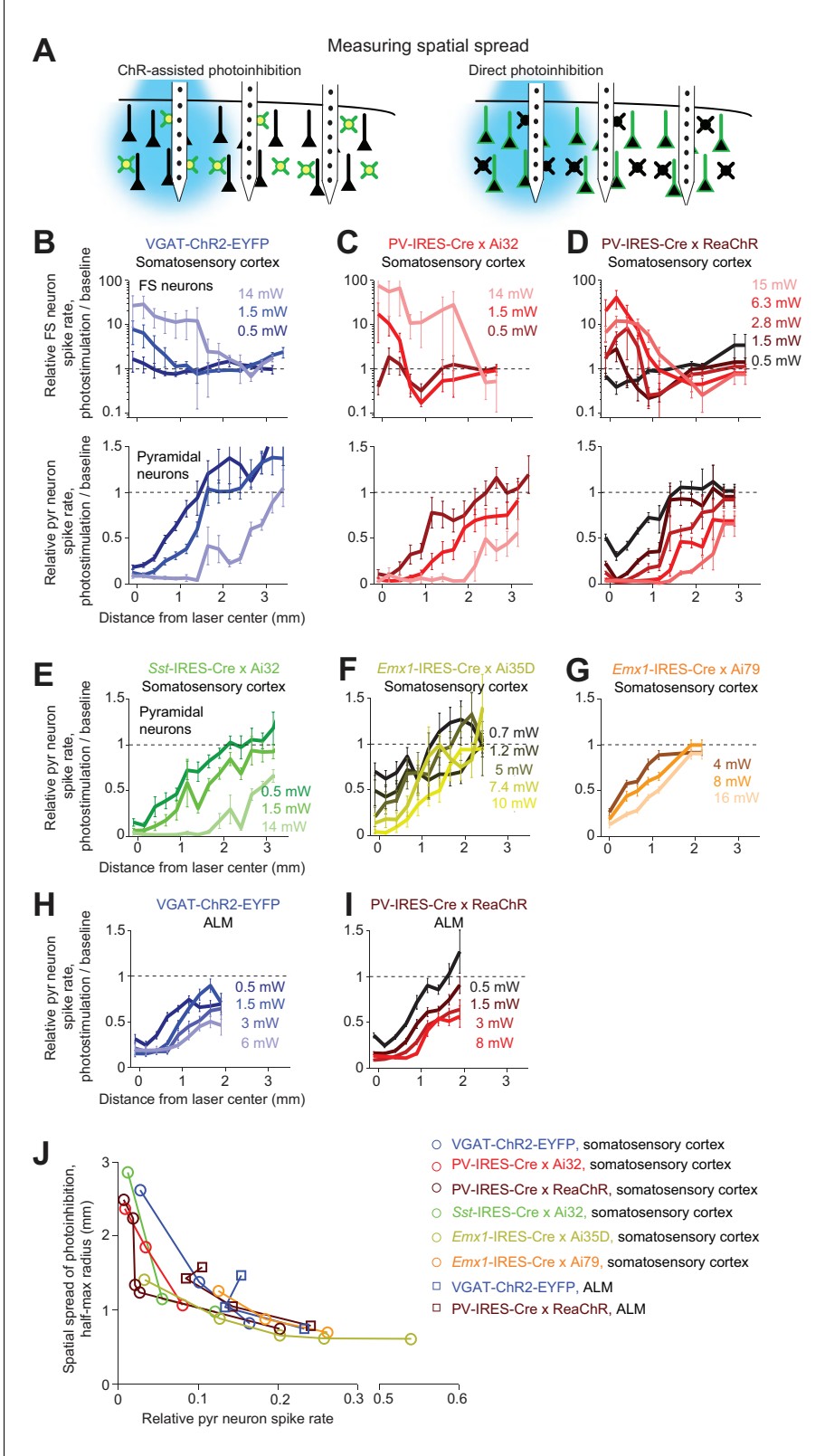

**Figure 5.** Spatial profile of photoinhibition. (**A**) Silicon probe recording at different lateral distances from photostimulus. (**B**) Relative spike rate versus lateral distance from the photostimulus center for three laser powers (0.5, 1.5, 14 mW). Blue light photostimulation in barrel cortex of VGAT-ChR2-EYFP mice. *Top*, FS neurons (n = 18). *Bottom*, pyramidal neurons (n = 111). Neurons were pooled across cortical depths. Mean ± s.e.m. across neurons, bootstrap (Materials and methods). (**C**) Same as (**B**), but for blue light photostimulation in barrel cortex of PV-IRES-Cre x Ai32 mice (FS neurons, n = 5;
*Figure 5 continued on next page*

*Figure 5 continued*

pyramidal neurons, n = 16). (D) Same as (B), but for orange light photostimulation in barrel cortex of PV-IRES-Cre x R26-CAG-LSL-ReaChR-mCitrine mice (FS neurons, n = 10; pyramidal neurons, n = 82). (E) Same as (B), but for blue light photostimulation in barrel cortex of *Sst*-IRES-Cre x Ai32 mice. Pyramidal neurons only (n = 65). (F) Same as (E), but for orange light photostimulation in barrel cortex of *Emx1*-IRES-Cre x Ai35D mice (n = 26). (G) Same as (E), but for orange light photostimulation in barrel cortex of *Emx1*-IRES-Cre X Camk2a-tTA X Ai79 mice (n = 174). (H) Same as (E), but for blue light photostimulation in ALM of VGAT-ChR2-EYFP mice (n = 96). (I) Same as (E), but for orange light photostimulation in ALM of PV-IRES-Cre x R26-CAG-LSL-ReaChR-mCitrine mice (n = 129). (J) Photoinhibition strength versus spatial spread. Relative spike rate is the average across all pyramidal neurons near laser center (<0.4 mm, all cortical depths). Spatial spread is the distance at which photoinhibition strength is half of that at the laser center ('radius, half-max'). Each circle represents data from one photostimulation power. Lines connect all circles of one method.

surrounding brain areas over more than a millimeter (*Figure 5J*). Considering the size of the mouse brain (10 mm long) this constraint on resolution is substantial. We looked for conditions that could break the relationship between photoinhibition strength and spatial spread. We tested the spatial spread of direct photoinhibition induced by GtACR1. In three different cortical regions, GtACR1-mediated photoinhibition also produced broad photoinhibition (*Figure 6A*), but the spatial spread was smaller than that produced by other photoinhibition methods (*Figure 6B*). For example, at 90% spike rate reduction near the laser center, the half-max radius of photoinhibition was 0.8 mm, compared to >1 mm for other methods (*Figure 6B*).

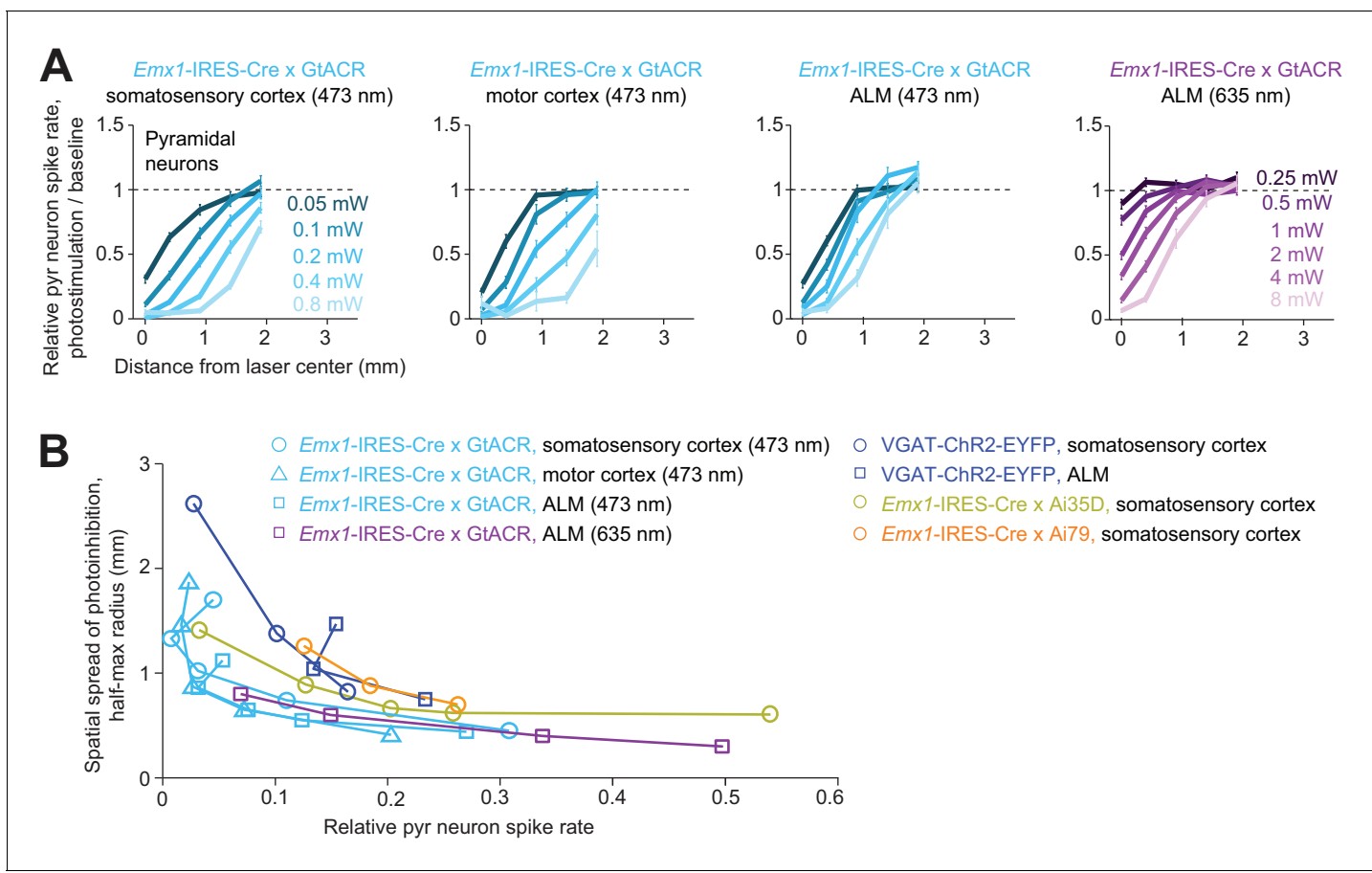

**Figure 6.** Spatial profile of direct photoinhibition in GtACR reporter mouse. (A) Relative spike rate versus lateral distance from the photostimulus center for various laser powers. Pyramidal neurons only. Blue light (437 nm) photostimulation in barrel cortex, n = 198; blue light photostimulation in motor cortex, n = 236; blue light photostimulation in ALM, n = 335; red light (635 nm) photostimulation in ALM, n = 236. Neurons were pooled across cortical depths. Mean ± s.e.m., bootstrap across neurons. (B) Photoinhibition strength versus spatial spread. Relative spike rate is the average across all pyramidal neurons near laser center (<0.4 mm, all cortical depths). Spatial spread is the distance at which photoinhibition strength is half of that at the laser center ('radius, half-max'). Each circle represents data from one photostimulation power. Lines connect all circles of one method. VGAT-ChR2-EYFP, *Emx1*-IRES-Cre x Ai35D, *Emx1*-IRES-Cre x Camk2a-tTA x Ai79, data from *Figure 5J* replotted here for reference.

We next sought to limit the spatial spread of ChR-assisted photoinhibition by directly limiting the spatial profile of interneuron activation. Transgenic mice express ChR in all GABAergic neurons. Even a small photostimulus activates interneurons over a larger volume due to light scattering (*Figures 4C* and *5B–D*). To limit activation of interneurons we injected small volumes of Cre-dependent ChR2 virus in PV-IRES-Cre mice (*Cardin et al., 2009*; *Lee et al., 2012*; *Lien and Scanziani, 2013*; *Pafundo et al., 2016*). The virus injection localized the expression of ChR2 (diameter of expression, 500 µm, *Figure 7A*). Silicon probe recordings (*Figure 7B*) confirmed that photostimulation excited FS neurons only at the infection site (*Figure 7C*). In the surrounding regions, FS neuron spike rates were even slightly suppressed (e.g. *Figure 7C*, at 400 µm away). Photoinhibition of pyramidal neurons extended additional several hundred micrometers, with neurons suppressed even 800 µm away from the photostimulus center. At 90% spike rate reduction at the infection site, the half-max radius of photoinhibition was approximately 0.5 mm (*Figure 7D*). Thus, limiting the spatial

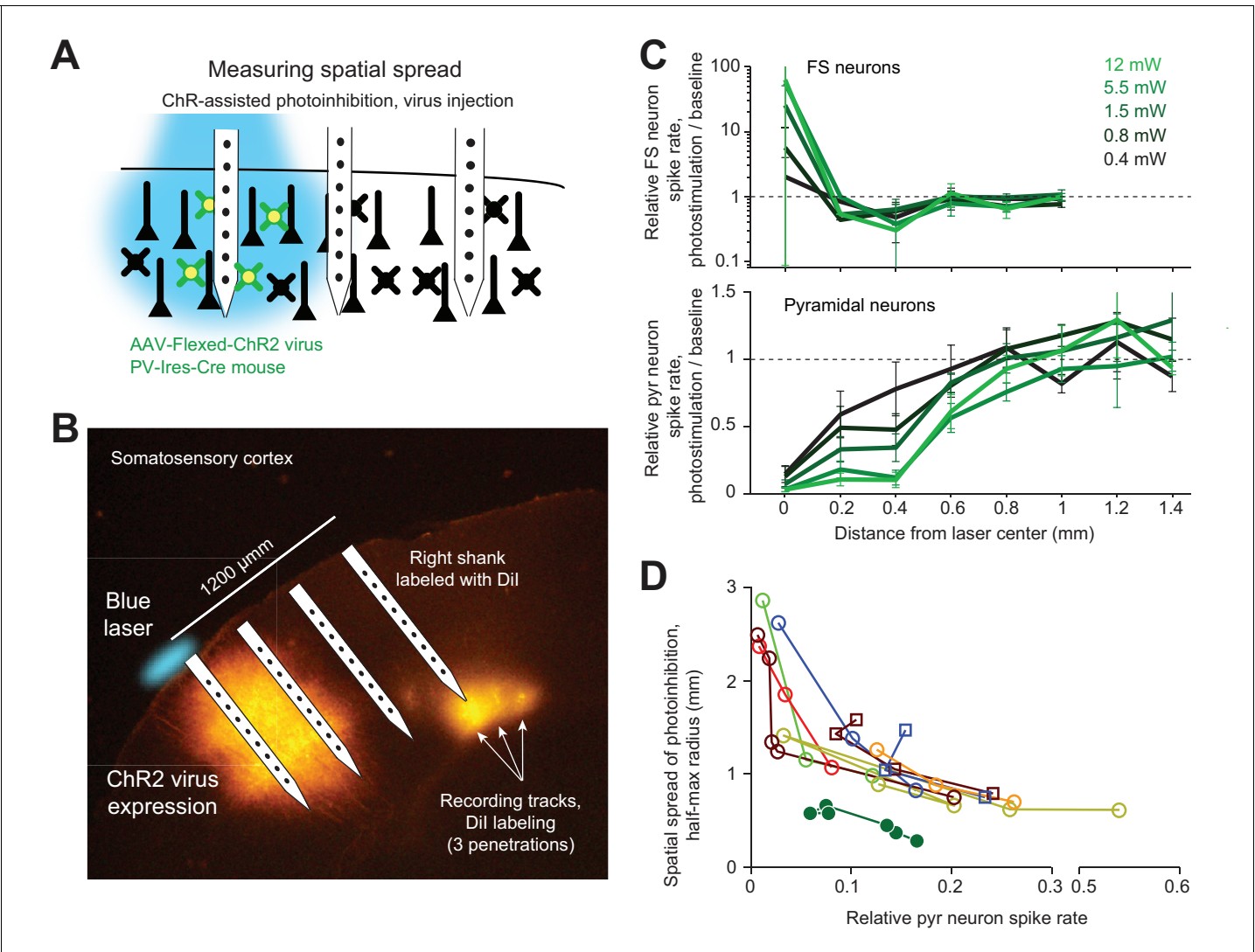

**Figure 7.** ChR-assisted photoinhibition using virus injection can achieve submillimeter spatial resolution. (A) Schematics, confined ChR2 expression in PV neurons and silicon probe recording at different distances from the expression site. (B) Silicon probe recording in barrel cortex during photostimulation. The right shank of the silicon probe was painted with DiI to label the recording tracks. Coronal section showing viral expression of ChR2-tdTomato, electrode and photostimulus locations. The photostimulus was aligned to the virus injection site. (C) Relative spike rate versus lateral distance from the photostimulus center for different laser powers. *Top*, putative FS neurons (n = 14). *Bottom*, pyramidal neurons (n = 78). Neurons were pooled across cortical depths. (D) Same as *Figure 5J*, but with virus injection data added.

extend of interneuron excitation reduced photoinhibition spread. Overall these results suggest that optgogenetic inactivation methods have a fundamental resolution on the order of 1 mm (*Figure 8*).

## Strong coupling between cortical neurons and the paradoxical effect

What underlies the spread of photoinhibition? Cortical neurons are coupled with each other. Activity reduction in a small region in the vicinity of the photostimulus withdraws input to the surrounding regions, reducing activity in the surround. Consistent with this interpretation, in regions surrounding the photostimulation site, a concurrent decrease in activity was observed in both FS neurons and pyramidal neurons, even with ChR-assisted photoinhibition (*Figure 9*, arrows). Activity decreased in proportion in FS and pyramidal neurons relative to their baseline activity.

In standard models of cortical circuits, networks are stabilized by inhibition to prevent runaway excitation (*Rubin et al., 2015*; *Tsodyks et al., 1997*; *van Vreeswijk and Sompolinsky, 1996*). These inhibition-stabilized networks (ISN) exhibit paradoxical effects. Selective excitation of interneurons paradoxically decreases interneuron activity together with excitatory neurons (*Litwin-Kumar et al., 2016*; *Pehlevan and Sompolinsky, 2014*; *Rubin et al., 2015*; *Sadeh et al., 2017*; *Tsodyks et al., 1997*). Because of recurrent excitatory connectivity in cortical circuits, the effects on excitatory neuron spike rates are amplified, which causes a large reduction in excitation of inhibitory interneurons.

We tested this prediction in the somatosensory cortex. In mice expressing ReaChR in PV neurons, we used orange light to illuminate a large part of the cortex (2 mm beam diameter at 4σ at the surface of the brain) to drive excitation in nearly all PV neurons across cortical layers (*Figure 10A*). Pyramidal neurons monotonically decreased their activity as a function of light intensity (*Figure 10B*). The majority of recorded excitatory neurons were in layer 5. Interestingly, FS neurons also decreased their activity, roughly in proportion despite being excited by ReaChR. As a function of light intensity, activity in FS neurons continued to decrease until most of the pyramidal neuron activity was silenced (*Figure 10B*, at 0.4 mW/mm²). At higher light intensities the FS neuron activity increased. This increase was likely driven by an increased photocurrent. In addition, the network also transitioned to a regime with little coupling due to lack of excitatory activity. These data are consistent with the

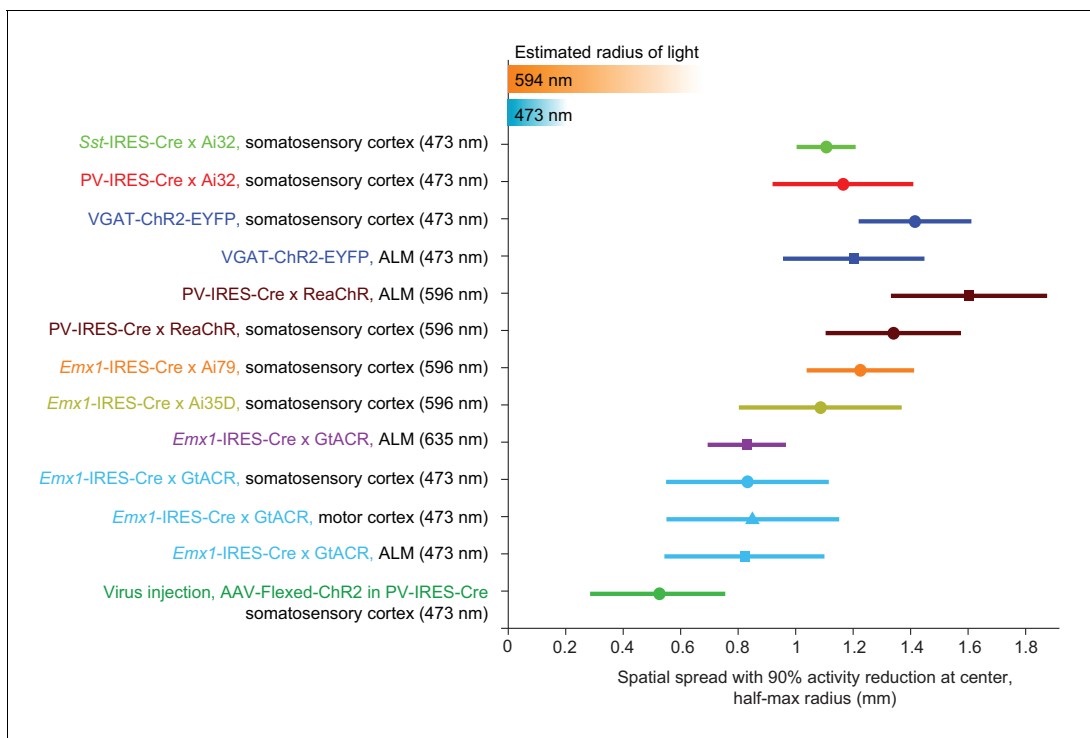

**Figure 8.** Summary of spatial resolution for all photoinhibition methods. Half-max radius of photoinhibition when the activity reduction at laser center is 90%. Data based on *Figures 5J*, *6B* and *7D* when relative pyramidal neuron spike rate is 0.1. Error bars show 90% confidence interval, bootstrap (Materials and methods). Estimated radius of light is based on data in *Figure 3D and G*.

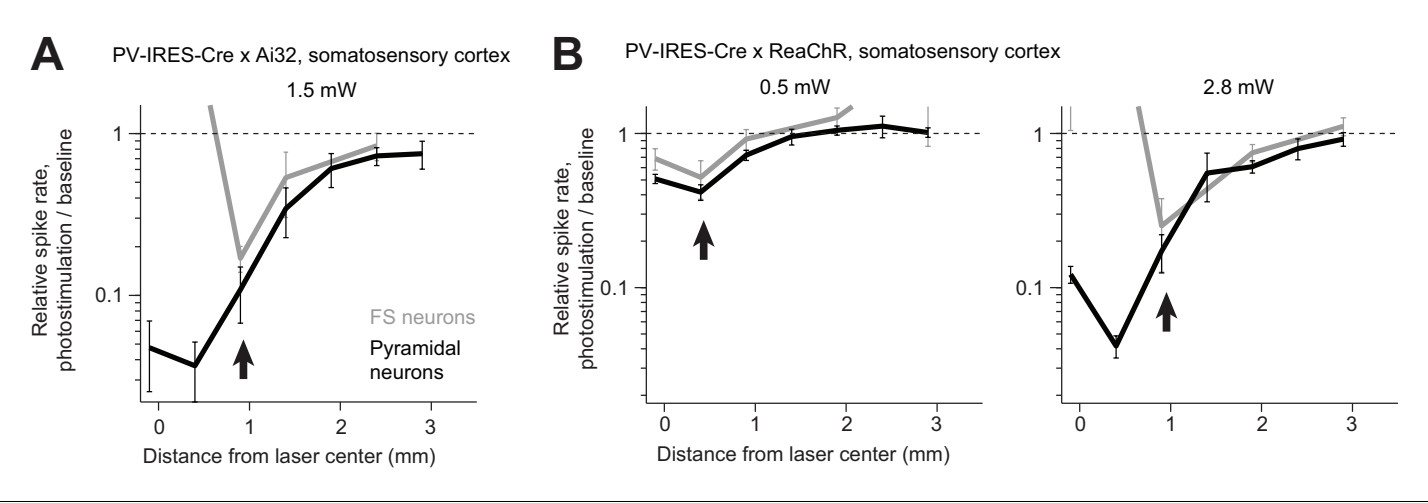

**Figure 9.** Proportional activity decrease in pyramidal and FS neurons during ChR-assisted photoinhibition. (**A**) Relative spike rate versus lateral distance from the photostimulus center for PV-IRES-Cre x Ai32. Data from *Figure 5C* replotted with activity shown on a log scale. FS neurons (gray) and pyramidal neurons (black). The arrows point to regions in the photostimulus surround where activity of FS neurons and pyramidal neurons decrease in proportion (paradoxical effect). (**B**) Same as (**A**) but for PV-IRES-Cre x ReaChR. Data from *Figure 5D*.

paradoxical effect predicted by cortical circuit models stabilized by inhibition (*Tsodyks et al., 1997*). However, the detailed dependence of the activity of excitatory and FS neurons as a function of the strength of the photostimulus remains to be explained.

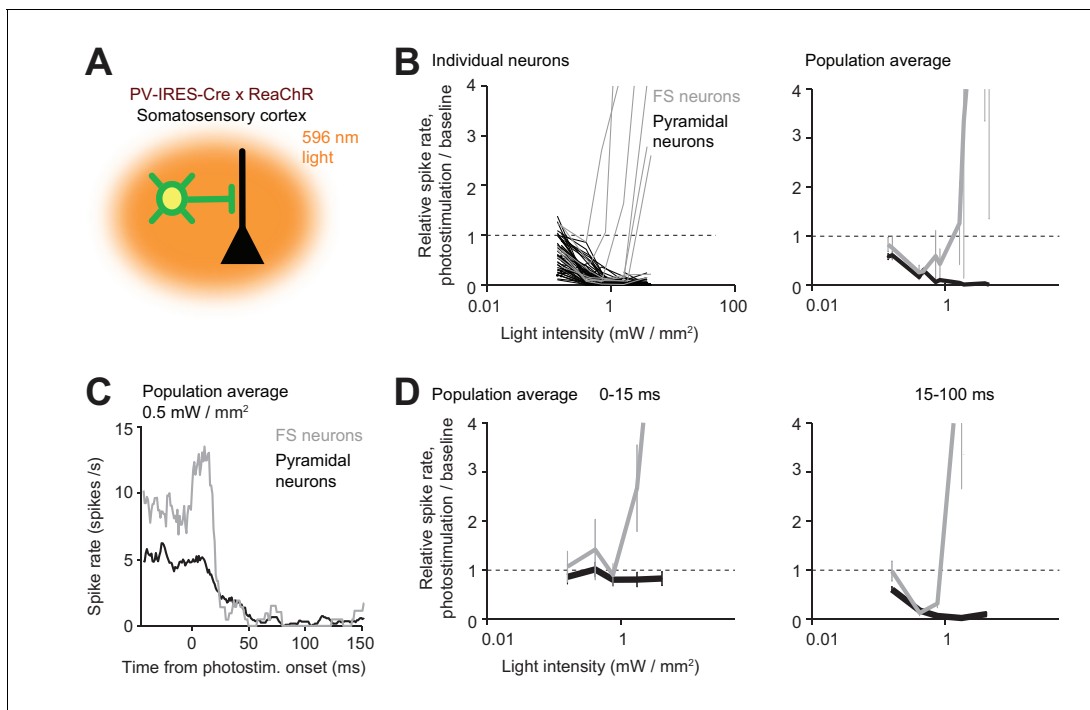

**Figure 10.** The paradoxical effect. (**A**) Photostimulating PV neurons using orange light. Laser beam diameter, 2 mm (Materials and methods). (**B**) Relative spike rate as a function of light intensity (<0.4 mm from laser center, all cortical depths). FS neurons (gray) and pyramidal neurons (black). *Left*, individual neurons (lines). *Right*, mean ± s.e.m. across neurons, bootstrap. Laser power was divided by the illuminated area to obtain light intensity. FS neurons, n = 10, pyramidal neurons n = 82. (**C**) Mean peristimulus time histogram for FS neurons (gray) and pyramidal neurons (black). (**D**) Same as (**B**) but for relative spike rate at different epochs of photostimulation.

## Temporal profile of optogenetic inactivation

We examined the dynamics of photoinhibition. For ChR-assisted photoinhibition, the kinetics of ChR determined the dynamics of the interneurons (*Figure 11A*). For example, in mice expressing ChR2 (VGAT-ChR2-EYFP), FS neuron activity was time-locked to the photostimulus. In mice expressing ReaChR (PV-IRES-Cre x ReaChR), which has slower off kinetics compared to ChR2 (*Lin et al., 2012*), FS neuron activity was not time-locked to the photostimulus and was strongly attenuated over prolonged photostimulation. Despite the different interneuron dynamics, the photoinhibition of pyramidal neurons was similarly (*Figure 11A*). A subset of FS neurons were excited throughout the photostimulation (*Figure 11B*, 0.8–1 s). Other FS neurons were suppressed by photostimulation on average; however, a subset of these inhibited FS neurons were transiently excited by the photostimulus, followed by inhibition (*Figure 11B*, compare 0–10 ms to 0.8–1 s), implying that these neurons were also expressing ChR. These data suggest that prolonged photostimulation (>20 ms) triggered spikes in a subset of FS interneurons, which in turn reduced activity in other FS neurons and pyramidal neurons.

For ChR-assisted photoinhibition, the photoinhibition lagged FS neuron excitation by 3 ms (Materials and methods; at >1.5 mW, FS excitation onset, $1.1 \pm 0.2$ ms). Photoinhibition was detectable $4.0 \pm 1.7$ ms after photostimulation onset (mean ± s.e.m., based on t-test of spike counts against baseline) and reached a maximum at $18.4 \pm 1.6$ ms ('photoinhibition onset', *Figure 11C*). ChR2-assisted photoinhibition has a slightly faster onset (14.6 ms) than ReachR-assisted photoinhibition (18.8 ms; p<0.001, bootstrap, one-tailed test, Materials and methods). Direct hyperpolarization of pyramidal neurons using Arch produced even more rapid photoinhibition onset (5 ms, *Figure 11C*).

Photoinhibition onset was progressively delayed at increasing distance from the laser center (*Figure 11D–E*). Photoinhibition at 2 mm away lagged the photoinhibition at the laser center by 10 ms (*Figure 11E*). This suggests a gradual spread of photoinhibition, consistent with the interpretation that coupling between cortical neurons mediated the photoinhibition spatial spread.

The temporal profile of photoinhibition offset was limited by rebound activity. Removal of photoinhibition elevated spike rate above baseline ('rebound activity'), which decayed to baseline over hundreds of milliseconds (*Figure 12A*). Rebound activity depended on both photostimulus duration and intensity (*Figure 12B*). For short photostimulation durations, rebound activity was moderate regardless of photostimulus intensity (*Figure 12C*, e.g. at 500 ms duration, rebound activity was 10% of baseline for 1.5–7 mW). For long photostimulation durations, rebound activity was substantial even at low laser power (*Figure 12C*, e.g. following a 4 s duration, rebound spike rate was >30% of baseline spike rate even at 1.5 mW). At moderate photostimulation duration (1 s), all photoinhibition methods produced some rebound activity regardless of photoinhibition strength (*Figure 12D*, Pearson's correlation, r = −0.2, p=0.29). The rebound activity could be caused by recovery from synaptic depression in silenced excitatory neurons, synaptic depression in activated FS neurons, or perturbed intracellular ion concentrations accumulated over prolonged duration of photostimulation (*Mahn et al., 2016*; *Wiegert et al., 2017*).

In summary, photoinhibition onset was rapid (10 ms), but the offset was limited by rebound activity. Attenuating the photostimulus gradually (100 ms) or reducing the photostimulus durations can both reduce rebound activity (*Figure 12*). These factors impose constraints on the temporal resolution of photoinhibition.

## Discussion

We examined optogenetic methods to locally silence neural activity in the mouse neocortex. ChR-assisted photoinhibition was more effective at lower light intensities (1.5 mW) in suppressing local pyramidal neurons than direct photoinhibition using the light-gated ion pumps Arch and Jaws. This power is unlikely to produce heating or other non-specific effects on neural activity (*Christie et al., 2012*; *Owen et al., 2019*; *Stujenske et al., 2015*). A soma-localized Cl- channel, GtACR1 (*Govorunova et al., 2015*; *Mahn et al., 2018*), produced the most potent silencing. However, transient excitation was induced over some range of laser power (0.3 mW or higher), which was likely due to axonal excitation, since soma targeting is not perfect. Potent silencing without excitation was still achieved over a large range of laser powers (0.05–0.3 mW). We generated a reporter mouse that expresses the soma-localized GtACR1 (*Mahn et al., 2018*) under the control of Cre-

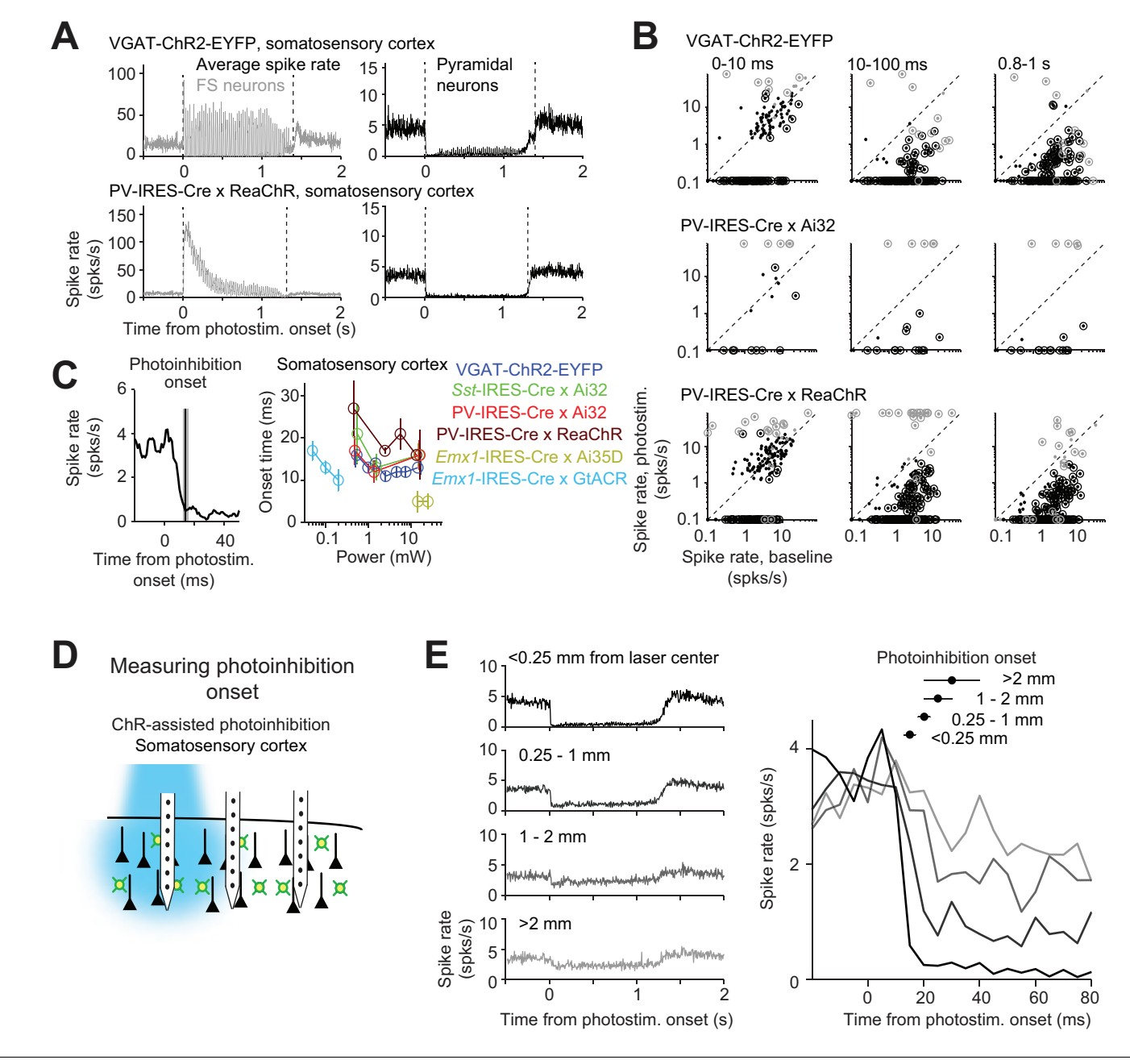

**Figure 11.** Time course of ChR2-assisted photoinhibition. (**A**) Mean peristimulus time histogram (PSTH, 1 ms bin) for FS neurons (gray) and pyramidal neurons (black). All neurons within 0.4 mm from the laser center across all cortical depths were pooled. Laser power, 14–15 mW. VGAT-ChR2-EYFP, FS neurons, n = 12, pyramidal neurons, n = 152; PV-IRES-Cre x ReaChR, FS neurons, n = 23, pyramidal neurons, n = 207. (**B**) Spike rate of FS neurons (gray) and pyramidal neurons (black) at different epochs of photostimulation. Dots correspond to individual neurons. Neurons with significant firing rate changes relative to baseline (p<0.05, two-tailed *t*-test) are highlighted by circles. (**C**) Photoinhibition onset time. *Left*, mean PSTH of pyramidal neurons in VGAT-ChR2-EYFP mice (1.5 mW) and photoinhibition onset (mean ± s.e.m.). The photoinhibition onset is the time when spike rate reached 90% of the average spike rate reduction during the whole photostimulation period. *Right*, photoinhibition onset for different methods. The color scheme of photoinhibition methods is the same as in *Figures 1E* and *2C*. Each circle represents data from one photostimulation power. Lines connect all circles of one method. (**D**) Schematic, measuring photoinhibition onset at different distances from the photostimulus. (**E**) *Left*, mean PSTH of pyramidal neurons at different distances from the photostimulus center. *Right*, spike rate at the onset of photostimulation (t = 0). Data from motor cortex and barrel cortex in VGAT-ChR2-EYFP, PV-IRES-Cre x Ai32, and PV-IRES-Cre x ReaChR mice are pooled (<0.25 mm, n = 301; 0.25–1 mm, n = 317; 1–2 mm, n = 262;>2 mm, n = 156). Laser power, 14–15 mW. Mean (± s.e.m. across neurons, bootstrap) photoinhibition onset latencies are shown on top.

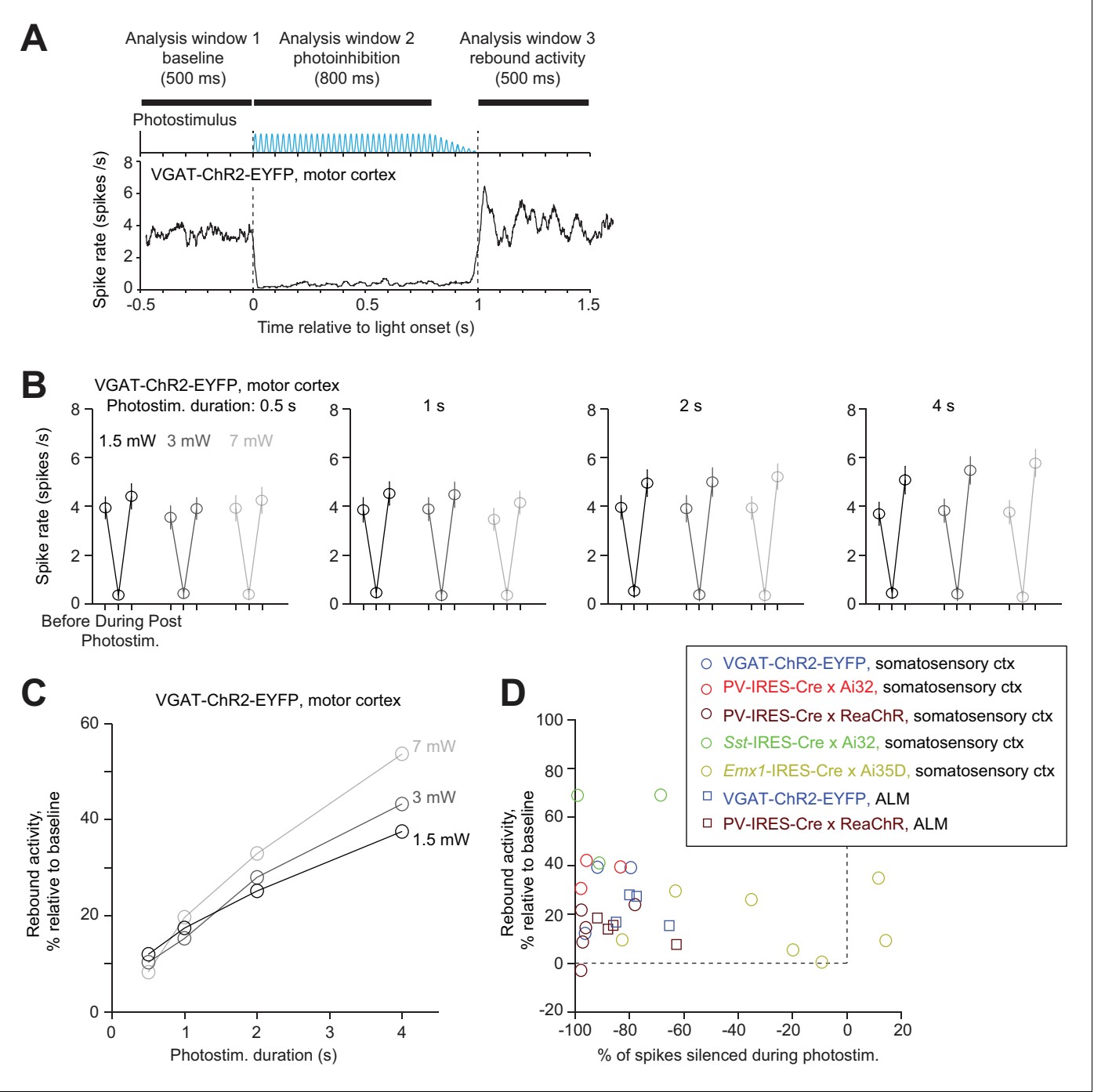

**Figure 12.** Rebound activity after photoinhibition. (**A**) Mean PSTH of pyramidal neurons during photoinhibition in VGAT-ChR2-EYFP mice. Dashed lines, photostimulus onset and offset. All neurons within 0.4 mm from the laser center across all cortical depths (n = 78). Data from motor cortex. Laser power, 7 mW. Spike rates were analyzed in three time windows before, during, and after photostimulation. Photostimulus duration, 1 s. (**B**) Spike rate of pyramidal neurons before, during, and after photostimulation for different laser powers and photostimulation durations. Mean ± s.e.m. across neurons, bootstrap. (**C**) Rebound activity as a function of laser powers and photostimulation durations. Relative activity is the after-photostimulation spike rate increase from the baseline, normalized to the baseline spike rate (Materials and methods). (**D**) Rebound activity as a function of photoinhibition strength. Percent of spikes silenced was relative to the baseline spike rate. Data from all mouse lines. VGAT-ChR2-EYFP, barrel cortex, n = 111; PV-IRES-Cre x Ai32, barrel cortex, n = 16; PV-IRES-Cre x ReaChR, barrel cortex, n = 82; *Sst*-IRES-Cre x Ai32, barrel cortex, n = 65; *Emx1*-IRES-Cre x Ai35D, barrel cortex, n = 26; VGAT-ChR2-EYFP, ALM, n = 96; PV-IRES-Cre x ReaChR, ALM, n = 129.

recombinases. This transgenic mouse is a useful tool to silence activity in genetically defined neuron populations.

Photobleaching measurements show that blue light is highly confined in cortical tissue. Comparison of the lateral spread of blue light, and the corresponding lateral spread of photoexcitation of FS neurons, suggests that the spread of light may not be the only determinant shaping the spatial aspects of the network response. This view is further confirmed by the observation that blue light and orange light produce similar spatial profiles of photoexcitation of FS neurons, despite the fact that orange light spreads much further than blue light. It is likely that interactions between the spread of light, the sizes of dendritic arbors, and the interplay between inhibition and recurrent excitation shape the response of neurons to light.

Our photobleaching measurements show that blue light delivered to the surface of the cortex results in an intensity distribution that decays rapidly as a function of depth in the cortex (*Figure 3*). Only a small proportion of light propagates below the superficial cortical layers. Consistent with this intensity distribution, at moderate laser powers (1.5 mW) photoinhibition in ChR2-EYFP mice was limited to cortex, with no obvious silencing in subcortical regions immediately below cortex (*Guo et al., 2014b*). However, higher light intensities in these mice can cause photoinhibition in the hippocampus and striatum beneath cortical photostimulation sites (*Babl et al., 2019*). Silencing mediated by red light is more likely to cause wide-spread direct optogenetic effects compared to blue light, since red light penetrates deeper into the tissue (*Lin et al., 2013*). These findings highlight the need for measurements of the spread of optogenetic effects. Depending on the application, it may be possible to limit off-target silencing by using transgenic mouse lines with restricted expression of Cre recombinase. For example, in our experiments expression of GTACR1 was limited to cortex using Emx1-Cre mice.

We characterized the spatial profile of different photoinhibition methods. Photoinhibition appeared uniform across the cortical depth and spread laterally over 1 mm or more, far beyond the spatial spread of light (0.25–0.5 mm). The spatial extent of photoinhibition was similar in somatosensory and motor cortex despite differences in microcircuits and connections with thalamus, and similar across photoinhibition methods (*Figures 5–8*). The spatial profiles of photoinhibition, laterally and along cortical layers, were also similar across different photostimulation wavelengths (473 vs. 594 nm), despite large differences in the spatial profiles of the photostimuli (i.e. light intensity) (*Figure 3*).

Local infection with Cre-dependent, ChR2-expressing virus in PV-IRES-Cre mice produced the smallest lateral spread in photoinhibition (radius, approximately 0.5 mm). Local expression of ChR limited the excitation of FS interneurons to a small region (~200 um in radius, *Figure 7C*). Additional lateral spread of photoinhibition was likely produced by the cortical circuit. Silencing activity in a focal region caused a withdrawal of excitatory input to surrounding cortical regions (*Kato et al., 2017*; *Lefort et al., 2009*; *Ozeki et al., 2009*; *Sanzeni, 2019*). For example, in regions surrounding the viral injection site, photostimulation caused a decrease in activity of both FS neurons and pyramidal neurons. Similar withdrawal of input could potentially explain the uniform photoinhibition across layers (*Hooks et al., 2011*; *Kiritani et al., 2012*), particularly for blue light mediated ChR-assisted photoinhibition (*Figure 4A*). Given the limited penetration of blue light in tissue at moderate laser powers (1.5 mW), the direct effects of photostimulation on neurons were most pronounced in L2/3 (*Figure 4C*), yet pronounced photoinhibition was observed in deep layers. Future studies directly recording inhibition onto neurons in the surround and deep layers could further clarify the underlying network effect.

Our measurements of the spatial profiles of inactivation were obtained within single brain regions (somatosensory or motor cortex). However the photostimuli were as far as 2.5 millimeters away and therefore often in different cortical areas. The structure of long-range feedforward and feedback connectivity likely shapes the spatial spread of inactivation. Feedforward projections target different layers compared to feedback projections (*Felleman and Van Essen, 1991*; *Mao et al., 2011*). It is also likely that the spatial spread of photoinhibition may be limited across boundaries between brain areas that are not coupled by long-range axons. Future experiments examining the coupling between brain regions contingent on long-range connectivity are still needed.

In both sensory and motor cortex, cortical neurons cannot maintain activity without thalamic drive (*Guo et al., 2017*; *Poulet et al., 2012*; *Reinhold et al., 2015*). Conversely, silencing activity in cortex reduces activity in the thalamus (*Guo et al., 2017*). Therefore, silencing a region in cortex likely

decreased activity in the parts of thalamus targeted by the silenced region. Since corticothalamic projections arise from the deep layers, and the direct effects of photostimulation is the strongest in superficial layers (particularly with blue light), our data imply that intracortical connections might also be required to maintain cortical activity, in addition to thalamocortical input. Resolving the relative contributions of intracortical connections and thalamocortical input in the maintenance of cortical activity will require additional experiments, guided by detailed circuit models (*Guo et al., 2017*; *Gutnisky et al., 2017*).

Our data provide additional evidence for strong coupling between cortical neurons. Over some range of photostimulus intensities, excitation of interneurons induced the 'paradoxical effect', where activity decreased in both interneurons and excitatory neurons (*Figure 10*). Excitation of GABAergic neurons reduces activity in nearby excitatory neurons, and thus reduced excitation to GABAergic neurons. Paradoxical effects were also observed in other experiments. For example, at the center of the laser, weak photostimulation (0.5 mW) of GABAergic neurons induced little increase or even a decrease in FS neurons activity, yet, pyramidal neuron activity was silenced (*Figure 9B*). These observations are consistent with predictions of inhibition-stabilized networks, and they suggest that cortical neurons operate in a regime with strong coupling (*Kato et al., 2017*; *Lefort et al., 2009*; *Ozeki et al., 2009*; *Pehlevan and Sompolinsky, 2014*; *Rubin et al., 2015*; *Tsodyks et al., 1997*; *Sanzeni, 2019*).

Previous studies manipulated cortical interneurons but did not find the paradoxical effect predicted by inhibition-stabilized network models (*Atallah et al., 2012*; *Gutnisky et al., 2017*; *Yu et al., 2016*). One key difference is that we weakly photostimulated nearly all PV neurons in superficial layers of transgenic mice. Previous studies used viral expression strategies that manipulated only subsets of PV neurons. Theoretical analysis and simulations suggest that the paradoxical effect is only induced when a large proportion of interneurons is excited (*Gutnisky et al., 2017*; *Sadeh et al., 2017*). Another key difference is photostimulation power. Cortical networks escape the paradoxical effect regime when most of the pyramidal neurons are silenced (*Figure 10B*), which effectively removes coupling between excitatory neurons. Consistent with this prediction, the paradoxical effect was only observed under weak photostimulation conditions that do not silence a majority of pyramidal neurons (*Figure 10*).

Our data provide guidance for the design of in vivo optogenetic experiments. All the data here were collected under awake non-behaving conditions, but our previous studies found little difference in photoinhibition strength across behavioral conditions (*Guo et al., 2014b*; *Li et al., 2016*). ChR-assisted photoinhibition and Cl- channel mediated directed photoinhibition are highly sensitive, producing similarly complete silencing near the laser center across a large range of laser powers (photoinhibition, 1.5–10 mW, *Figure 1E*). However, we note that the spatial spread of photoinhibition differed substantially across laser powers (*Figures 5* and *6*). Higher laser power removed spikes over a larger cortical region. This power-dependent spatial spread produces graded loss-of-function manipulations in vivo. For example, photoinhibiting the barrel cortex induced increasingly larger behavioral effects with increasing laser power across a large power range (1.5–10 mW) (*Guo et al., 2014b*).

Photoinhibition has been used extensively in loss-of-function studies to localize functions to specific brain regions. In the neocortex, the 1 mm length scale of photoinhibition poses a limit on the spatial resolution of loss-of-function manipulations. Interpretation of photoinhibition effects must take this spatial spread function into account. For example, silencing a brain region nearby to an involved region could produce false-positive behavioral effects. One work-around could involve systematically mapping the photoinhibition effects (e.g. a behavioral effect) around a region of interest, then deconvolve the known spatial spread function to recover the underlying region involved in behavior (*Li et al., 2016*).

Although we explored several spatiotemporal stimuli, the possible parameter space for photostimulus design is still sparsely explored. It is possible that sub-millimeter resolution of photoinhibition could be achieved with direct photoinhibition using more sensitive optogenetic effectors and spatially patterned illuminations. In addition, the spatial resolution of photoinhibition could differ across brain regions. In many subcortical regions GABAergic neurons make long-range projections, reducing the spatial resolution of ChR2-assisted photoinhibition. On the other hand, recurrent and lateral excitation is less pronounced in many subcortical regions compared to neocortex, promising better spatial resolution when direct photoinhibition is used for photoinhibition.

Photoinhibition has been used to probe the involvement of brain regions during specific behavioral epochs (*Guo et al., 2014b*; *Hanks et al., 2015*; *Li et al., 2015*; *Sachidhanandam et al., 2013*). Our results suggest that the temporal resolution of photoinhibition is limited by rebound activity (*Figure 12*), which could potentially produce confounding behavioral consequences. Rebound activity could be partially alleviated by changes in photostimulus parameters (*Figure 12C*) (*Wiegert et al., 2017*).

In many experiments, cell-type-specific manipulations produced multi-phasic network responses both on local and downstream circuits over hundreds of milliseconds (*Guo et al., 2017*). These results highlight that optogenetic circuit manipulations provide the most insights into functions of specific neural dynamics when the primary effects of the perturbation and, equally importantly, the effects on downstream brain areas are taken into account, ideally measured simultaneously in behaving animals (*Gao et al., 2018*; *Guo et al., 2017*; *Inagaki et al., 2019*; *Li et al., 2016*).

# Materials and methods

## Animals

This study is based on data from 45 mice (age >P60, both male and female mice). 35 transgenic mice were used to characterize photoinhibition, including 14 VGAT-ChR2-EYFP mice, 2 PV-IRES-Cre x Ai32 mice, 3 *Sst*-IRES-Cre x Ai32 mice, 11 PV-IRES-Cre x R26-CAG-LSL-ReaChR-mCitrine mice, 1 *Emx1*-IRES-Cre x Ai35D mouse, 2 *Emx1*-IRES-Cre x Camk2a-tTa x Ai79 mice, and 2 *Emx1*-IRES-Cre x R26-CAG-LNL-GtACR1-ts-FRed-Kv2.1 mice. 5 PV-IRES-Cre mice were used to drive photoinhibition using Cre-dependent ChR2 virus injections. 1 *Emx1*-IRES-Cre x Rosa26-LSL-H2B-mCherry mouse and 1 *Emx1*-IRES-Cre x Rosa-CAG-LSL-H2B-GFP mouse (gift from Josh Huang, Cold Spring Harbor Laboratory) were used for photobleaching experiments.

All procedures were in accordance with protocols approved by the Institutional Animal Care and Use Committees at Baylor College of Medicine (protocol AN7012) and Janelia Research Campus (protocol 14–115). All surgical procedures were carried out aseptically under 1–2% isoflurane anesthesia. Buprenorphine (0.5 mg/kg) or Sustained Release Buprenorphine (1 mg/kg) was used for pre- and post-operative analgesia. Ketoprofen (5 mg/kg) or Sustained Release Meloxicam (4 mg/kg) was used at the time of surgery and post-operatively to reduce inflammation.

## Generation of transgenic GtACR mice

Targeting vector, Rosa26-CAG-LNL-GtACR1-ts-FusionRed-Kv2.1C-WPRE-polyA, was derived from AAV.CamKIIa.GtACR1-ts-FRed-Kv2.1.WPRE (*Mahn et al., 2018*) and a standard Rosa26 backbone (*Dymecki and Kim, 2007*; *Madisen et al., 2012*; *Soriano, 1999*). The targeting vector DNA was electroporated into ES cells and the chimeric mice were generated using successfully targeted ES cell clones. GtACR1 expression was evaluated both by native fluorescence and functional assays. Rosa26-CAG-LNL-GtACR1-ts-FRed-Kv2.1 mice were submitted to The Jackson Laboratory (stock #033089).

## Surgery

Mice were prepared for photostimulation and electrophysiology with a clear-skull cap (*Guo et al., 2014b*) and a headpost (*Guo et al., 2014a*). The scalp and periosteum over the dorsal surface of the skull were removed. A layer of cyanoacrylate adhesive (Krazy glue, Elmer's Products Inc) was directly applied to the intact skull. A custom made headbar was placed on the skull (approximately over visual cortex) and cemented in place with clear dental acrylic (Lang Dental Jet Repair Acrylic; Part# 1223-clear). A thin layer of clear dental acrylic was applied over the cyanoacrylate adhesive covering the entire exposed skull, followed by a thin layer of clear nail polish (Electron Microscopy Sciences, Part# 72180).

In some PV-IRES-Cre mice ChR2 was introduced by injecting 50 nL of AAV2/5-hSyn1-FLEX-hChR2-tdTomato (Addgene plasmid 41015, Janelia Molecular Biology Shared Resource) (*O'Connor et al., 2013*) into the barrel cortex (bregma posterior 1.5 mm, lateral 3.5 mm) at two depths (400 and 800 μm), followed by implantation of the headbar. The injection was made through the thinned skull using a custom, piston-based, volumetric injection system. Glass pipettes (Drummond) were pulled and bevelled to a sharp tip (outer diameter of 30 μm) (*Petreanu et al., 2009*).

Pipettes were back-filled with mineral oil and front-loaded with viral suspension immediately before injection.

For silicon probe recordings, a small craniotomy was made over the recording site in mice already implanted with the clear-skull cap and headpost (see *Electrophysiology*). The dental acrylic and bone were thinned using a dental drill. The remaining thinned bone was carefully removed using a bent forceps. A separate, smaller craniotomy (diameter, approximately 600 μm) was made through the headpost bar for ground wires.

## Photostimulation

Light from a 473 nm laser (DHOM-M-473–200, UltraLaser) or a 594 nm laser (Cobolt Inc, Colbolt Mambo 100) or a 635 nm laser (MRL-III-635L-100mW, Changchun New Industries Optoelectronics Technology) was controlled by an acousto-optic modulator (AOM; MTS110-A3-VIS, Quanta Tech; extinction ratio 1:2000; 1μs rise time) and a shutter (Vincent Associates), coupled to a 2D scanning galvo system (GVSM002, Thorlabs), then focused onto the brain surface (*Guo et al., 2014b*). The laser at the brain surface had a Gaussian profile with a beam diameter of 400 μm at 4σ. We tested photoinhibition in barrel cortex (bregma posterior 1.5 mm, 3.5 mm lateral), anterior lateral motor cortex (ALM, bregma anterior 2.5 mm, 1.5 mm lateral) and primary motor cortex (bregma anterior 0.5 mm, 1.5 mm lateral). Photoinhibition was similar across different regions.

To prevent the mice from detecting the photostimulus, a 'masking flash' (40 × 1 ms pulses at 10 Hz) was delivered using a LED driver (Mightex, SLA-1200–2) and 470 nm or 590 nm or 627 nm LEDs (Luxeon Star). The masking flash began before the photostimulus started and continued through the end of the epoch in which photostimulation could occur.

The standard photostimulus had a near sinusoidal temporal profile (40 Hz) with a linear attenuation in intensity over the last 100–200 ms (duration: 1.3 s including the ramp). This temporal profile was chosen to minimize rebound activity based on pilot experiments. In some cases, we also used a constant photostimulus without ramp (*Figure 1F*). For photostimulation in *Emx1*-IRES-Cre x Camk2a-tTa x Ai79 mice, the photostimulus duration was 400 ms, including a 100 ms ramp. In the experiments presented in *Figure 12* we also tested other photostimulus durations (0.5, 1, 2, and 4 s including a 200 ms ramp). The photostimuli were delivered at approximately 7 s intervals. The power and locations of photostimulation were chosen randomly. Because we used a time-varying photostimuli the power values reported here reflect the time-averaged power.

For photostimulation experiments testing the paradoxical effect (*Figure 10*), the laser at the brain surface had a diameter of 2 mm at 4σ. We tested paradoxical effect in barrel cortex (bregma posterior 1.5 mm, 3.5 mm lateral). To calculate light intensity ($mW/mm^2$) reported in *Figure 10*, laser power was divided by the area of the laser beam: light intensity = $power/2\pi r^2$, where r = 1 mm.

## Photobleaching

We measured photobleaching in vivo to characterize the spatial profile of light intensity at two different wavelengths (473 nm and 594 nm, *Figure 3*). In transgenic mice expressing GFP (for 473 nm) or mCherry (for 594 nm) in the nuclei of cortical pyramidal neurons, photobleaching was induced by a stationary beam and prolonged (10 min) illumination at different laser powers through the clear-skull cap. Nuclear fluorescence was imaged in fixed tissue sections using a confocal microscope (Zeiss LSM 510). Multiple sections were imaged around each photobleaching site.

GFP or mCherry fluorescence was measured in regions of interest (ROIs) corresponding to individual nuclei (*Figure 3A*). For each ROI, we calculated a ΔF by subtracting a baseline ROI intensity, $F_0$, from the mean fluorescence. $F_0$ was computed as the average intensity of all ROIs > 700 μm away from the laser center. For red (594 nm) light, photobleaching was relatively complete close to the center of the photostimulus (*Figure 3F*); we therefore used only ROIs with a lateral distance of > 700 μm to compute F0. For low amount of bleaching (-ΔF/ $F_0$<<1), ΔF/ $F_0$=$f(r)$ = - $k$ $I(r)$ $T$, where $I(r)$ is the light intensity in the tissue, and $T$ is the duration of illumination. The constant $k$ is related to the bleaching cross section of the fluorescent molecule.

For best contrast the bleaching close to the center of the laser beam was greater than 50%. We therefore measured ΔF/ $F_0$ near the laser center (within 100 μm for GFP, 400 μm for mCherry) for different light doses (*Figure 3C*). The relationship between light dose and bleaching in general is exponential, $f(r)$ = $-1$ +exp(- $k$ $I(r)$ $T$). The distribution of light intensity was thus computed as $I(r)$ = -

constant * ln(f(r) + 1). The constant k was derived from empirical data (*Figure 3C*), with separate fits for GFP (473 nm light) and mCherry (594 nm light). *Figure 3D and G* show the distribution of light derived from f(r) at 200 mW x min light dose. We picked this power because of the high contrast (i.e. large $\Delta F/ F_0$ near the laser center). Similar spatial profiles were obtained at lower light doses.

## Electrophysiology

All recordings were carried out while mice were awake but non-behaving. Extracellular spiking activity was recorded using silicon probes. We used NeuroNexus probes with four shanks (at 200 or 400 μm spacing) and recording sites spaced 100 or 200 μm apart (eight sites per shank, P/N A4 × 8–5 mm-100-200-177, A4 × 8–5 mm-100-200-413, A4 × 8–5 mm-200-200-177, A4 × 8–5 mm-200-200-413, and, A4 × 8–5 mm-100-400-177). Silicon probes were connected to a headstage (Intan Technology) that multiplexed the 32-channel voltage recording into two analog signals (fabricated at Janelia Farm Research Campus, Brian Barbarits, Tim Harris). The multiplexed analog signals were recorded on a PCI6133 board at 312.5 kHz (National instrument) and digitized at 14 bit. The signals were demultiplexed into the 32 voltage traces at the sampling frequency of 19531.25 Hz and stored for offline analyses in a custom software spikeGL (C. Culianu, Anthony Leonardo, Janelia Farm Research Campus). For recordings in *Emx1*-IRES-Cre x R26-CAG-LNL-GtACR1-ts-FRed-Kv2.1 mice, we used 64-channel Cambridge NeuroTech silicon probes (H2 acute probe, 250 μm spacing, 2 shanks). The 64-channel voltage signals were multiplexed, recorded on a PCI6133 board (National Instrument) and digitized at 400 kHz (14 bit). The signals were demultiplexed into 64 voltage traces sampled at 25 kHz and stored for offline analysis. The headstage was mounted on a motorized micromanipulator (MP-285, Sutter Instrument).

A 1 mm diameter craniotomy over the recording site was made prior to the recording sessions. In PV-IRES-Cre mice prepared for ChR2 virus mediated photoinhibition, a larger craniotomy (2 mm x 1 mm) was made around the injection site. The position of the craniotomy was guided by stereotactic coordinates for recordings in ALM (bregma anterior 2.5 mm, 1.5 mm lateral), motor cortex (bregma anterior 0.5 mm, 1.5 mm lateral), or barrel cortex (bregma posterior 1.5 mm, 3.5 mm lateral).

Prior to each recording session, the tips of the silicon probe were brushed with DiI in ethanol solution and allowed to dry. The surface of the craniotomy was kept moist with saline. The silicon probe was positioned on the surface of the cortex and advanced manually into the brain at ~3 μm/s, normal to the pial surface. The depth of the electrode tip ranged from 749 to 1000 μm below the pial surface. The electrode depth was inferred from manipulator depth and verified with histology. To minimize pulsation of the brain, a drop of silicone gel (3–4680, Dow Corning, Midland, MI) was applied over the craniotomy after the electrode was in position. The tissue was allowed to settle for several minutes before the recording was started.

## Histology

After the conclusion of recording experiments, mice were perfused transcardially with PBS followed by 4% PFA/0.1 M PB. The brains were fixed overnight and sectioned. Coronal sections (100 μm) were cut and images of DiI labeled recording tracks were acquired on a macroscope (Olympus MVX10). Electrode tracks were compared to the manipulator depth readings (*Guo et al., 2014b*). In *Figure 4*, the definition of cortical layer boundaries in barrel cortex was based on mouse brain atlas (*O'Connor et al., 2010*).

## Data analysis

The extracellular recording traces were band-pass filtered (300–6 kHz). Events that exceeded an amplitude threshold (four standard deviations of the background) were subjected to manual spike sorting to extract single-units (*Guo et al., 2014b*).

Our final data set comprised of 2638 single units (barrel cortex, 1158; ALM, 1105; M1, 375). For each unit, its spike width was computed as the trough to peak interval in the mean spike waveform (*Figure 1B*). We defined units with spike width <0.35 ms as FS neurons (366/2638) and units with spike width >0.45 ms as putative pyramidal neurons (2206/2638). Units with intermediate values (0.35–0.45 ms, 36/2638) were excluded from our analyses.

To quantify the strength of inactivation, we derived a 'relative spike rate'. Relative spike rate was computed for either individual neurons (*Figures 1D*, *4C* and *10B*) or for the entire neuronal

populations. For each neuron, we computed its spike rate during the photostimulus and its baseline spike rate (500 ms time window before photostimulus onset). For relative spike rate of individual neurons, each neuron's spike rate with photostimulation was normalized by dividing its baseline spike rate. For relative spike rate of the neuronal population, the spike rates with photostimulation were first averaged across the population (without normalization) and then normalized by dividing the averaged baseline spike rate. The 'relative spike rate' reports the total fraction of spiking output under photostimulation.

To quantify the lateral spread of inactivation, we computed relative spike rate at different distances from the photostimulus, and the distance at which inactivation is half of max, that is its strength at the photostimulus center (*Figure 5J*, 'radius, half-max'). In *Figure 8*, we quantified the half-max radius of photoinhibition when the relative spike rate is 0.1 at laser center (i.e. 90% activity reduction). Bootstrap was performed over neurons to establish 90% confidence intervals around the spatial spread estimates in *Figure 8*. For each round of bootstrapping, repeated 2000 times, we randomly sampled with replacement neurons in the dataset. We computed the half-max radius on the resampled datasets. The bootstrap created a distribution of half-max radius estimates. The confidence interval in *Figure 8* encompassed 90% of the observed half-max radiuses from bootstrap.

To quantify the time course of inactivation, we computed the average population Peristimulus Time Histogram (PSTH) aligned to photostimulus onset (*Figure 11*). To quantify the onset time of inactivation, we estimated the time when the pyramidal neuron PSTH reached its minimum post stimulus onset ($18.4 \pm 1.6$ ms, mean $\pm$ s.e.m.). To quantify the earliest time inactivation occurred ($3.95 \pm 1.7$ ms), we estimated the first time bin (10 ms bin) in which the spike rate differed significantly ($p<0.05$ two-tailed t-test) from baseline. To quantify photoinhibition lag from excitation of the interneurons, we computed the earliest time bin in which FS neuron PSTH significantly deviated from baseline spike rate ($1.05 \pm 0.2$ ms) and subtracted this onset time from the photoinhibition onset time (lag: 2.90 ms).

Bootstrap was performed over neurons to obtain standard error of the mean. For each round of bootstrapping, repeated 1000–10000 times, we randomly sampled with replacement neurons in the dataset. We computed the means of the resampled datasets. The standard error of the mean was the standard deviation of the mean estimates from bootstrap.

Bootstrap was also used to evaluate whether ChR2-assisted photoinhibition had significantly faster onset latency than ReaChR-assisted photoinhibition (*Figure 11C*). The neuronal dataset was re-sampled with replacement, and the onset latencies were recalculated from the resampled datasets. The p-value reported is the fraction of times when bootstrap produced a resampled dataset in which ReaChR-assisted photoinhibition had a faster onset latency than ChR2-assisted photoinhibition, (one-tailed test against the null hypothesis that ChR2-assisted photoinhibition had significantly faster onset latency).

To quantify rebound activity, for each neuron, we computed spike rate in a 500 ms window after photostimulus offset, subtract the baseline spike rate (computed in a 500 ms time window before photostimulus onset), and normalized this spike rate difference to the baseline spike rate. The rebound activity (i.e. activity increase from baseline) was quantified as the percentage of baseline spike rate. The rebound activity was averaged across the recorded population (*Figure 12C–D*).

## Statistics

The electrophysiology sample sizes are similar to sample sizes used in the field: more than 100 units per brain region. No statistical methods were used to determine sample size. All key results were replicated in multiple mice. Mice were allocated into experimental groups according to strain and brain regions tested. Unless stated otherwise, the investigators were not blinded to allocation during experiments and outcome assessment. During spike sorting, experimenters cannot tell photostimulation conditions, so experimenters were blind to conditions. Statistical comparisons using t-tests and bootstrap are described in detail in sections above.

## Data and code availability

Rosa26-CAG-LNL-GtACR1-ts-FRed-Kv2.1 mice are available at The Jackson Laboratory (stock #033089). Electrophysiology data and code used are available at Github (https://github.com/

NuoBCM/PhotoinhibitionCharaterization; copy archived at https://github.com/elifesciences-publications/PhotoinhibitionCharaterization).

## Acknowledgements

We thank Mathias Mahn and Ofer Yizhar for sharing the GtACR1-ts-FRed-Kv2.1 construct. We thank Mingshan Xue for comments on the manuscript, David Golomb for critical discussions, Susan Michael and Amy Hu for histology, Tim Harris, Brian Barbarits, Wei-Lung Sun, Anthony Leonardo for help with silicon probe recordings. This work was supported by Howard Hughes Medical Institute (KS), Helen Hay Whitney Foundation fellowship (NL, HI), Sir Henry Wellcome Postdoctoral Fellowship (SC), the Robert and Janice McNair Foundation (NL), Whitehall Foundation (NL), Alfred P Sloan Foundation (NL), Searle Scholars Program (NL), NIH NS104781 (NL), the Pew Charitable Trusts (NL), and Simons Collaboration on the Global Brain (#543005, KS, NL).

## Additional information

### Competing interests

Karel Svoboda: Reviewing editor, *eLife*. The other authors declare that no competing interests exist.

### Funding

| Funder | Grant reference number | Author |
|---|---|---|
| Howard Hughes Medical Institute | | Karel Svoboda |
| Simons Foundation | Simons Collaboration on the Global Brain (#543005) | Nuo Li Karel Svoboda |
| Helen Hay Whitney Foundation | | Nuo Li Hidehiko K Inagaki |
| Wellcome | | Susu Chen |
| Robert and Janice McNair Foundation | | Nuo Li |
| Whitehall Foundation | | Nuo Li |
| Alfred P. Sloan Foundation | | Nuo Li |
| Kinship Foundation | | Nuo Li |
| National Institutes of Health | NS104781 | Nuo Li |
| Pew Charitable Trusts | | Nuo Li |

The funders had no role in study design, data collection and interpretation, or the decision to submit the work for publication.

### Author contributions

Nuo Li, Conceptualization, Data curation, Formal analysis, Supervision, Funding acquisition, Validation, Investigation, Writing—original draft, Writing—review and editing; Susu Chen, Hidehiko K Inagaki, Resources, Data curation, Formal analysis, Validation, Investigation, Methodology, Writing—review and editing; Zengcai V Guo, Han Chen, Yan Huo, Data curation, Formal analysis, Validation, Investigation, Writing—review and editing; Guang Chen, Data curation, Formal analysis, Investigation; Courtney Davis, Investigation; David Hansel, Methodology, Writing—review and editing; Caiying Guo, Resources; Karel Svoboda, Conceptualization, Resources, Data curation, Formal analysis, Supervision, Funding acquisition, Validation, Methodology, Writing—original draft, Writing—review and editing

## Author ORCIDs
Nuo Li https://orcid.org/0000-0002-6613-5018
David Hansel http://orcid.org/0000-0002-1352-6592
Karel Svoboda https://orcid.org/0000-0002-6670-7362

## Ethics
Animal experimentation: All procedures were in accordance with protocols approved by the Institutional Animal Care and Use Committees at Baylor College of Medicine (protocol AN7012), Janelia Research Campus (protocol 14-115).

## Decision letter and Author response
Decision letter https://doi.org/10.7554/eLife.48622.sa1
Author response https://doi.org/10.7554/eLife.48622.sa2

## Additional files

### Supplementary files
• Transparent reporting form

### Data availability
Rosa26-CAG-LNL-GtACR1-ts-FRed-Kv2.1 mice are available at The Jackson Laboratory (stock #033089). Electrophysiology data and code used are available at Github (https://github.com/NuoBCM/PhotoinhibitionCharaterization; copy archived at https://github.com/elifesciences-publications/PhotoinhibitionCharaterization).

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
