## [Decision Letter]

**Acceptance summary:**

Li et al. examine various methods for optogenetic inhibition of specific brain regions. Optogenetic methods for precisely inactivating cortical areas are of central importance to mouse systems neuroscience. The authors utilize novel quantitative methods to study of the spatiotemporal effects of optogenetic cortical inactivation, and find, somewhat surprisingly that a wide variety of optical and optogenetic approaches all produce near complete silencing of all cortical layers with a lateral resolution of approximately one mm. This paper will be a valuable resource for design and interpretation of studies performing cortical silencing.

**Decision letter after peer review:**

Thank you for submitting your article "Spatiotemporal limits of optogenetic manipulations in cortical circuits" for consideration by *eLife*. Your article has been reviewed by four peer reviewers, one of whom is a member of our Board of Reviewing Editors, and the evaluation has been overseen Eve Marder as the Senior Editor. The following individual involved in review of your submission has agreed to reveal their identity: Carl CH Petersen (Reviewer #2).

The reviewers have discussed the reviews with one another and the Reviewing Editor has drafted this decision to help you prepare a revised submission.

Summary:

Perturbations are an essential tool for understanding causal mechanisms in the brain, and focal inactivation of cortex with optogenetics is a useful paradigm for assessing localization of cortical function. However, due to recurrent circuitry, perturbations produce downstream effects that need to be quantified and understood. In this study, Li et al. systematically benchmark the spatial and temporal limits of optogenetic silencing of the cortex using a broad range of current tools. They test tools using two strategies: 1) ChR2 assisted-photoinhibition, which leverages inhibitory interneurons to suppress cortical pyramidal cells, and 2) direct photo-inhibition of pyramidal cells.

The authors find, perhaps surprisingly, that all methods produce a similar spatial extent of cortical suppression, on the order of ~1mm. Of considerable note, this photo-inhibition profile extends beyond the spatial profile of light delivery, suggesting resolution is limited by the structure of the cortical circuitry.

The authors also generate and characterize a new transgenic line that expresses a light-gated silencing (anion) channel (GtACR). This channel is tagged with a somatic localization sequence to minimize axonal expression. Cortical silencing with GtACR was the most potent with effects at very low light levels.

The results provide practical guidance for the design and interpretation of optogenetic silencing in the mouse neocortex.

Essential revisions:

1) The paper is not written very clearly. Please improve the presentation including experimental approach, interpretation and logic. Please correct language, grammar, and syntax.

2) There are frustrating aspects of the study, especially in terms of what was *not* studied. For example, the assays of inactivation do not adequately explore a complete dynamic range of light intensities. In fact, for many of the manipulations, the minimal power used was apparently nearly maximally effective (e.g. Figure 1E, most opsins), and so the reader is left with incomplete knowledge of the power levels actually needed to silence neural networks. This is critical to know given Kreitzer lab's recent report on non-specific light induced artefacts in the brain.

3) One of the main findings, that the primary effects of photon delivery mainly extend to superficial cortex, and yet this leads to global inactivation across all cortical layers is extremely important (see point 5a, below), and may explain why it doesn't matter which optogenetic approach is used – the results are generally the same. It is important to get this message out. However, the speculation in the Discussion on the role for thalamocortical interactions in propagating the silencing between layers is just that -- speculation. This is one area where addition experiments to test this hypothesis would be extremely useful, although not required.

4) In many places the approach and rationale for the data analysis was opaque. a) It is not clear how the authors have quantified their "relative spike rates" in most figures. In the Materials and methods it states: "The spike rates with photostimulation were averaged across the population and normalized by dividing the averaged baseline spike rate." Does this mean the baseline spike rates are from the averaged population but not for individual neurons? What was the rationale for this, rather than just normalizing each neuron to its own spike rate? Is this also true for figures where individual neurons are plotted (e.g. Figure 1D)?

b) In many figures, the legend states that the error bars are SEM from bootstrap, but there is limited information on the bootstrapping in the Materials and methods. What was being bootstrapped (different subsamples of neurons for the population average and baseline response?), and how many bootstraps were performed? Also what is the rational for using SEM (which presumably depends on the number of bootstraps) instead of the confidence interval?

c) More information is needed to understand the photobleaching experiment. It wasn't clear how the "empirical relationship" between laser power and photobleaching was computed. Is this the exponential fit in Figure 1C? Is the data in Figure 3D and G from a specific light power- or is the distribution for all light powers, just varying in magnitude? Does this process account for potential differences in light absorption (and photobleaching sensitivity) of GFP and mCherry? That is, how do we know that the difference in effect between the different wavelengths of light is due to the wavelength or the fluorophore? It is also unfortunate that the light powers used for this experiment do not match the ones used for the physiology- in fact, 5 mW (the lowest light power used) is two orders of magnitude higher than the light power that is sufficient to locally suppress the cortex to ~25% of initial rates with Emx1-Cre x GtACR (Figure 6). As such, it is hard to understand how the measurements made reflect the conditions that are actually used- or if this method is sensitive enough to accurately measure the true distribution of light intensity. Clarifying these relationships is important since the authors use these measurements later to make arguments about the mechanism through which suppression propagates in the cortex (see Major Concern 2a).

d) How were layers defined in laminar recordings?

5) In many cases, the authors make definitive statements where it is not entirely clear where the supporting data is or whether that is the only possible interpretation. A number of examples are below, but there are likely others.

a) The authors make a strong argument that the direct effects of photoinhibition are restricted to the superficial layers and drive the photoinhibition in deep layers through indirect effects. They seem to base this conclusion on two observations: 1) the lack of penetration of blue light from their photobleaching experiments and 2) the restriction of excitation of FS cells to superficial layers. Concerns about the interpretation of the photobleaching experiment are in point 1c. In addition, the authors' conclusions about the effects on FS cells are limited by cell numbers, their ability to accurately identify ChR2 expressing neurons, and the contribution of paradoxical effects. Most importantly, their own data is not entirely consistent with this conclusion: if the deep layer suppression is mainly caused by the loss of activity in superficial layers, one would not observe the striking difference in the suppression in layer 5/6 given the similar degree of suppression in superficial layers with different light powers in Figure 4.

b) The authors indicated that the paradoxical effects they observed were mainly due to reduced excitation from nearby excitatory neurons to GABAergic neurons (subsection “Strong coupling between cortical neurons and the paradoxical effect”). However, the paradoxical effects could also be induced by increased inhibition from activation of nearby GABAergic neurons. The authors also state that the increase in FS neuron activity at higher light intensities is "partly driven by increased photocurrent". This seems likely to be true, but what evidence to the authors have for this statement? And why only "partly"- what else contributes?

c) The statement regarding the observation of "axonal excitation" was unexpected and it wasn't clear why the authors concluded that the increase in firing rates were due to activation of the axonal compartment. Also- are Figures 2C and D the same data, just at different time scales?

d) In describing Figure 11C, the authors state that "… the photoinhibition lagged FS neuron excitation by 3 ms…". However, there is no clear quantification of the onset of FS neuron activation.

e) The authors should be clear that given their cell-type identification is only dependent on the spike width, that the classifications are "putative". There is plenty of evidence for narrow spiking excitatory cells and broad spiking interneurons.

6) The authors need to provide some explanation for why the laminar profile of suppression is so similar with blue and red light using EMX1-cre GtACR, given their expected differences in the tissue penetration properties. Similarly, the authors should provide some explanation for why the spatial spread of photoinhibition is smaller with viral injections than transgenic expression if blue light spread is not a limitation.

7) The authors need to provide some discussion of the fact that all of their measurements were made under conditions of spontaneous activity. Should we expect all of these methods to be similarly effective at silencing areas the animal is receiving sensory input or actively engaged in behavior tasks?

8) The title of the paper "Spatiotemporal limits of optogenetic manipulations in cortical circuits" is somewhat misleading in that it is clearly possible to carry out optogenetic manipulations on a finer scale than carried out here, indeed even in some cases with single-cell resolution.

9) Figure legends need to be greatly improved. It should be possible to understand what is going on in all parts of the figure. Each panel/subgraph should at least be explained.

10) How worried should users of the GtACR reporter line be about axonal activation? Cortex can be silenced with light intensities of 0.1-0.2 mW, but axons begin spiking around 0.8 mW. Are users safe if they restrict intensities < 0.3 mW? Are there tell-tale signs of axon activation that users should lookout for, or are there specific controls that would always be recommended?

[Editors' note: further revisions were requested prior to acceptance, as described below.]

Thank you for resubmitting your work entitled "Spatiotemporal constraints on optogenetic inactivation in cortical circuits" for further consideration at *eLife*. Your revised article has been favorably evaluated by Eve Marder as the Senior Editor, a Reviewing Editor, and two reviewers.

The manuscript has been improved but there are some remaining issues that need to be addressed before acceptance, as outlined below:

While the revised manuscript is substantially improved in its clarity of presentation, there still remain a number of issues that have not been satisfactorily addressed.

1) One major strengths of the paper is that it quantitively assesses both light delivery into the brain and downstream effects of light. The innovative approach of using photobleaching to assess light penetration is a reasonable one, but far from perfect, in that it requires much greater levels of light, and/or over extended periods of time compared to optogenetic activation/inhibition. Accordingly, there is a level of imprecision to the method that should be much more explicitly addressed in the paper, either in explaining the limitations are in providing additional data.

Critically, the spread of light does indeed appear to depend on the light power used. Indeed, the authors seem to suggest this themselves when responding to point 6 ("At high laser power the light illuminates a larger volume"). This is also very evident in Figure 3B. However, the photobleaching data suggest that the penetration of the light is independent of light power (or light dose, as the authors now call it), and thus argue that the higher powers they use only confer better signal to noise. It seems like some independent validation/calibration of this approach is needed.

2) The authors state in their rebuttal that the calculation of light intensity does not depend on differences across fluorophores. However, the k in their equation seems to be fluorophore dependent. More information is needed on whether the same k was used across fluorophores, or whether this was part of their fitting process, and if so whether these values are in line with expectations about the fluorophores. In addition, some citations that support which factors do and don't contribute to the photobleaching process would be helpful.

3) If the radius of laser light scatter is ~200 microns, then it is still confusing why the viral injection, which has a radius of ~250 microns (i.e. greater than the light) should restrict the spread of inhibition compared to in the transgenic condition. This result suggests that the authors are underestimating the spread of their light.

4) The terminology regarding light levels in the revised manuscript and rebuttal letter is quite inconsistent, with moderate meaning different things (1 mW or 20 mW) in different contexts. This terminology should be standardized throughout the paper, especially in relation to light levels required for biological effects.

5) It would be valuable to add some discussion regarding how these optogenetic silencing approaches seem to be extremely sensitive such that there is not much dynamic range for the manipulation, regarding e.g. the possibility to explore the behavioral consequences of different degrees of cortical suppression. On a related note, the new limited data provided in the rebuttal on silencing with low level light delivery should be included in the revised paper.

---

## [Author Response]

Essential revisions:1) The paper is not written very clearly. Please improve the presentation including experimental approach, interpretation and logic. Please correct language, grammar, and syntax.

We have reworked the text with clarity in mind.

2) There are frustrating aspects of the study, especially in terms of what was not studied. For example, the assays of inactivation do not adequately explore a complete dynamic range of light intensities. In fact, for many of the manipulations, the minimal power used was apparently nearly maximally effective (e.g. Figure 1E, most opsins), and so the reader is left with incomplete knowledge of the power levels actually needed to silence neural networks. This is critical to know given Kreitzer lab's recent report on non-specific light induced artefacts in the brain.

For Arch-mediated suppression, we covered a large dynamic range. For ChR-assisted inhibition, the photoinhibition was already strong at the lowest powers used, 0.5 mW. We now include additional data for ReaChR x PV-cre mice collected over a larger range of powers down to 0.3 mW, which induced partial inactivation (Author response image 1). The dose-response is very similar for blue light photostimulation in VGAT-ChR2-EYFP mice (see VGAT-ChR2-EYFP characterization data at low laser power in Inagaki et al., 2019, Extended Data Figure 7). For ChR-assisted methods, the dose-response is similar across methods. For example, 80% activity suppression at 0.5 mW, and 50% activity suppression at 0.3 mW.

The typical powers used in this study are below the powers used in many optogenetic studies. For example, the Kreitzer study (Owen et al., 2019) investigate the non-specific effect of light stimulation at 532 nm using laser power in the 3-15 mW range. The most notable effect on neural activity in their study was induced at 15 mW. Most power levels in our experiments were well below that, 0.5 mW-10 mW (Figure 1E). We have revised the text to clarify this (Results, and Discussion).

For the new soma-targeted GtACR reporter mouse, the power needed for photoinhibition is even lower, i.e. 0.1 mW for >80% suppression, which alleviate concerns related to heating and other light-induced effects.

**Author response image 1. respfig1:** Normalized spike rate as a function of laser power. Pyramidal neurons from ALM (< 0.4 mm from laser center, all cortical depths). Spike rates were normalized to baseline (dashed line, see Materials and methods). Mean ± s.e.m. across neurons, bootstrap. PV-IRES-Cre x ReaChR, 55 neurons.

3) One of the main findings, that the primary effects of photon delivery mainly extend to superficial cortex, and yet this leads to global inactivation across all cortical layers is extremely important (see point 5a, below), and may explain why it doesn't matter which optogenetic approach is used – the results are generally the same. It is important to get this message out. However, the speculation in the Discussion on the role for thalamocortical interactions in propagating the silencing between layers is just that -- speculation. This is one area where addition experiments to test this hypothesis would be extremely useful, although not required.

We have sought to further elucidate the underlying mechanism by examining photoinhibition onset latency across layers. If photoinhibition in the deep layers resulted indirectly from thalamo-cortical loops, the photoinhibition should take effect slower in the deep layers. However, we did not observe any significant differences: above 500 µm depth, latency=15 ms, n=50 neurons; below 500 µm, latency=15 ms, n=119 (blue light photostimulation, VGAT-ChR2-EYFP and PV-cre x Ai32 data combined). However, this negative result could be due to low statistical power. We acknowledge that our existing data cannot elucidate the causes of laminar spread. We have revised the text to reflect this (Discussion).

Because cortical and thalamic activity are interdependent (Guo/Inagaki et al., 2017), disambiguating the contribution of intracortical vs. thalamo-cortical connections could be challenging in vivo. We welcome good suggestions.

4) In many places the approach and rationale for the data analysis was opaque. a) It is not clear how the authors have quantified their "relative spike rates" in most figures. In the Materials and methods it states: "The spike rates with photostimulation were averaged across the population and normalized by dividing the averaged baseline spike rate." Does this mean the baseline spike rates are from the averaged population but not for individual neurons? What was the rationale for this, rather than just normalizing each neuron to its own spike rate? Is this also true for figures where individual neurons are plotted (e.g. Figure 1D)?

We have clarified the description in the Results and Materials and methods. We measured the light sensitivity of different optogenetic inactivation methods by computing the 'relative spike rate', which is the spike rate of putative pyramidal neurons during inactivation, averaged across neurons, divided by the baseline spike rate. Neurons with low spike rate contributes less to the population relative spike rate.

b) In many figures, the legend states that the error bars are SEM from bootstrap, but there is limited information on the bootstrapping in the Materials and methods. What was being bootstrapped (different subsamples of neurons for the population average and baseline response?), and how many bootstraps were performed? Also what is the rational for using SEM (which presumably depends on the number of bootstraps) instead of the confidence interval?

We have clarified the description in the Materials and methods. The bootstrap analysis is nonparametric, and the estimated SEM reflects the confidence interval around the observed mean given the variability across neurons.

c) More information is needed to understand the photobleaching experiment. It wasn't clear how the "empirical relationship" between laser power and photobleaching was computed. Is this the exponential fit in Figure 1C? Is the data in Figure 3D and G from a specific light power- or is the distribution for all light powers, just varying in magnitude? Does this process account for potential differences in light absorption (and photobleaching sensitivity) of GFP and mCherry? That is, how do we know that the difference in effect between the different wavelengths of light is due to the wavelength or the fluorophore? It is also unfortunate that the light powers used for this experiment do not match the ones used for the physiology- in fact, 5 mW (the lowest light power used) is two orders of magnitude higher than the light power that is sufficient to locally suppress the cortex to ~25% of initial rates with Emx1-Cre x GtACR (Figure 6). As such, it is hard to understand how the measurements made reflect the conditions that are actually used- or if this method is sensitive enough to accurately measure the true distribution of light intensity. Clarifying these relationships is important since the authors use these measurements later to make arguments about the mechanism through which suppression propagates in the cortex (see concern 2a).

We have clarified the Materials and methods to include this information:

1) The empirical relationship refers to the exponential fit in Figure 3C, which converts any ∆F/F0 value to a power level.

2) Figure 3D and G are derived from the 20 mW data in Figure 3B. We picked this intermediate light dose because it showed the largest contrast (large ΔF/ F0) without fully bleaching nuclei near the laser center, to avoid floor effects. The estimated spatial profile was robust to this choice. Below we show spatial profiles derived using data at two other light doses (Author response image 2). All light doses produced similar results.

3) Photobleaching is proportional to the total light dose (i.e. intensity x time). High laser powers and long durations are used to obtain sufficient contrast in ΔF/ F0. The calculation does not depend on absorption differences across fluorophores. It also does not depend on the absolute power level used. The spatial profiles in Figure 3 show the fractional light intensity relative to the laser center. The same profile applies to any power level and the same spatial profile can be derived using photobleaching data obtained at any light dose (Author response image 2).

**Author response image 2. respfig2:** The estimated spatial profile of blue light in tissue. *Top*, Light intensity profile derived from changes in fluorescence (ΔF/ F0) at three different light doses. *Bottom*, light intensity is shown as a function of cortical depth (left) and lateral distance (right) from the laser center. Profiles derived from photobleaching data at three different light doses are shown in different colors.

d) How were layers defined in laminar recordings?

We have clarified the description in the Materials and methods.

5) In many cases, the authors make definitive statements where it is not entirely clear where the supporting data is or whether that is the only possible interpretation. A number of examples are below, but there are likely others.a) The authors make a strong argument that the direct effects of photoinhibition are restricted to the superficial layers and drive the photoinhibition in deep layers through indirect effects. They seem to base this conclusion on two observations: 1) the lack of penetration of blue light from their photobleaching experiments and 2) the restriction of excitation of FS cells to superficial layers. Concerns about the interpretation of the photobleaching experiment are in point 1c. In addition, the authors' conclusions about the effects on FS cells are limited by cell numbers, their ability to accurately identify ChR2 expressing neurons, and the contribution of paradoxical effects. Most importantly, their own data is not entirely consistent with this conclusion: if the deep layer suppression is mainly caused by the loss of activity in superficial layers, one would not observe the striking difference in the suppression in layer 5/6 given the similar degree of suppression in superficial layers with different light powers in Figure 4.

Our current data do not elucidate the circuit mechanism of laminar spread. Also see response to point 3 above. For ChR2-assisted photoinhibition at weak laser power (0.5 and 1.5 mW, Figure 4B), the direct effect of photostimulation is the strongest in the superficial layers. Given the limited penetration of blue light, photoinhibition in the deep layers is likely caused by circuit effect. At high laser power, light spread can directly cause photoinhibition in the deep layers. For example, at 14 mW, FS neurons were activated in Layer 5 (Figure 4F). We revised the text in several places to more appropriately state these results.

“At moderate laser power (1.5 mW), blue light excited FS neurons mainly in superficial layers (Figure 4E-F). Yet, loss of activity was present in both superficial and deep layers.”

“Similar withdrawal of input could potentially explain the uniform photoinhibition across layers (Hooks et al., 2011; Kiritani et al., 2012), particularly for blue light mediated ChR-assisted photoinhibition (Figure 4B). […] Future studies directly recording inhibition onto neurons in the surround and deep layers could further clarify the underlying network effect.”

b) The authors indicated that the paradoxical effects they observed were mainly due to reduced excitation from nearby excitatory neurons to GABAergic neurons (subsection “Strong coupling between cortical neurons and the paradoxical effect”). However, the paradoxical effects could also be induced by increased inhibition from activation of nearby GABAergic neurons. The authors also state that the increase in FS neuron activity at higher light intensities is "partly driven by increased photocurrent". This seems likely to be true, but what evidence to the authors have for this statement? And why only "partly"- what else contributes?

In inhibition stabilized network models (ISN), the network suppression is driven by withdraw of excitatory input. However, extracellular recording cannot distinguish withdraw of excitation from increased inhibition. Future studies recording excitatory and inhibitory currents could distinguish these possibilities. Previous whole cell recordings have found that suppression of both excitatory and inhibitory synaptic inputs underlies surround suppression (Ozeki et al., 2009; Kato et al., 2017), which is a prediction of ISN. The observed paradoxical effect here is also consistent with predictions of ISN. We have removed the word “partly”.

c) The statement regarding the observation of "axonal excitation" was unexpected and it wasn't clear why the authors concluded that the increase in firing rates were due to activation of the axonal compartment. Also- are Figures 2C and D the same data, just at different time scales?

This is inferred to be axonal activation based on previous studies which report similar activation (Mahn et al., 2018; Messier et al., 2018). We have revised the text. “At higher laser powers transient, short-latency excitation was apparent, which was likely due to axonal excitation.”

Figure 2C and D show the same data. We now indicate this in the revised legends.

d) In describing Figure 11C, the authors state that "… the photoinhibition lagged FS neuron excitation by 3 ms…". However, there is no clear quantification of the onset of FS neuron activation.

This information was in the Materials and methods:

“To quantify photoinhibition lag from excitation of the interneurons, we computed the earliest time bin in which FS neuron PSTH significantly deviated from baseline spike rate (1.05 ± 0.2 ms), and subtracted this onset time from the photoinhibition onset time (lag: 2.90 ms).”

We now include this in the main text.

e) The authors should be clear that given their cell-type identification is only dependent on the spike width, that the classifications are "putative". There is plenty of evidence for narrow spiking excitatory cells and broad spiking interneurons.

We now state: “Units with narrow spikes were putative fast spiking (FS) neurons and likely expressed parvalbumin.”

6) The authors need to provide some explanation for why the laminar profile of suppression is so similar with blue and red light using EMX1-cre GtACR, given their expected differences in the tissue penetration properties. Similarly, the authors should provide some explanation for why the spatial spread of photoinhibition is smaller with viral injections than transgenic expression if blue light spread is not a limitation.

See response to point 3 and 5a. Our current data do not fully elucidate the circuit mechanism of laminar spread. In addition, GtACR1 is expressed in the dendrites of deep layer neurons, which extend into superficial layers. This could also contribute to the laminar spread in EMX1-cre X GtACR-Kv2.1 mice.

Photoinhibition is more restricted with viral expression of ChR2 than transgenic expression. At high laser power, the light illuminates a larger volume. For transgenic expression, high laser power excites ChR2 expressing interneurons far away from the laser center (Figure 5B, top). For viral expression, the excitation of the ChR2 expressing interneurons is spatially restricted (Figure 7C, top). This reasoning is now included in the text (subsection “Spatial profile of light intensity”).

7) The authors need to provide some discussion of the fact that all of their measurements were made under conditions of spontaneous activity. Should we expect all of these methods to be similarly effective at silencing areas the animal is receiving sensory input or actively engaged in behavior tasks?

We have previously compared photoinhibition under behaving and non-behaving conditions and found little difference in photoinhibition strength (Guo et al., 2014, Figure 2E; Li et al., 2016, Extended Data Figure 2B-D). We have included this information in the seventh paragraph of the Discussion.

8) The title of the paper "Spatiotemporal limits of optogenetic manipulations in cortical circuits" is somewhat misleading in that it is clearly possible to carry out optogenetic manipulations on a finer scale than carried out here, indeed even in some cases with single-cell resolution.

OK. We change the title to “Spatiotemporal constraints on optogenetic inactivation in cortical circuits”

9) Figure legends need to be greatly improved. It should be possible to understand what is going on in all parts of the figure. Each panel/subgraph should at least be explained.

We have improved the figure legends for clarity.

10) How worried should users of the GtACR reporter line be about axonal activation? Cortex can be silenced with light intensities of 0.1-0.2 mW, but axons begin spiking around 0.8 mW. Are users safe if they restrict intensities < 0.3 mW? Are there tell-tale signs of axon activation that users should lookout for, or are there specific controls that would always be recommended?

Below 0.3 mW, we find little axonal excitation measured by extracellular spiking. Axonal excitation begins to appear at 0.3 mW and is clear at 0.8 mW. We highlight this further in the Discussion. We recommend using low amount of light, but more importantly, we suggest directly measuring the effect of photostimulation on neural activity under the specific experimental conditions.

[Editors' note: further revisions were requested prior to acceptance, as described below.]The manuscript has been improved but there are some remaining issues that need to be addressed before acceptance, as outlined below:While the revised manuscript is substantially improved in its clarity of presentation, there still remain a number of issues that have not been satisfactorily addressed.1) One major strengths of the paper is that it quantitively assesses both light delivery into the brain and downstream effects of light. The innovative approach of using photobleaching to assess light penetration is a reasonable one, but far from perfect, in that it requires much greater levels of light, and/or over extended periods of time compared to optogenetic activation/inhibition. Accordingly, there is a level of imprecision to the method that should be much more explicitly addressed in the paper, either in explaining the limitations are in providing additional data.Critically, the spread of light does indeed appear to depend on the light power used. Indeed, the authors seem to suggest this themselves when responding to point 6 ("At high laser power the light illuminates a larger volume"). This is also very evident in Figure 3B. However, the photobleaching data suggest that the penetration of the light is independent of light power (or light dose, as the authors now call it), and thus argue that the higher powers they use only confer better signal to noise. It seems like some independent validation/calibration of this approach is needed.

We don't understand the reviewer's concern. At the low light intensities used here, scattering and absorption are linear. There is a fixed probability of photobleaching per absorbed photon. Therefore photobleaching is also linear. Because photobleaching cannot go below DF/F = -1 (i.e. complete bleaching) the profile of photobleaching broadens with light dose (Figure 3B). This does NOT mean that the spread of light broadens with intensity; everything is scaled up linearly as the illumination power is increased. In fact, given our calibration experiments we can calculate the normalized light intensity ('density of photons') and this is independent of light dose, as expected (Figure 3D, G). The relevant calculation is described explicitly in the Materials and methods.

We now include a discussion on limitations in the Results and Discussion sections. The photobleaching analysis was restricted to a limited dynamic range: at high light intensity, the region near the laser center is fully bleached, while at low light intensity, the bleaching is not visible. Therefore, the photobleaching assay required light doses that were far higher than those used in optogenetic experiments. Also due to the limited dynamic range, the photobleaching may not be sensitive enough to detect very low levels of light in the surround. As a result, our measurement likely misses a long tail in light intensity that is still able to activate light gated channels.

**Author response image 3. respfig3:** Conversion of the bleaching profile (∆F/F0) to light dose and normalized light intensity.

2) The authors state in their rebuttal that the calculation of light intensity does not depend on differences across fluorophores. However, the k in their equation seems to be fluorophore dependent. More information is needed on whether the same k was used across fluorophores, or whether this was part of their fitting process, and if so whether these values are in line with expectations about the fluorophores. In addition, some citations that support which factors do and don't contribute to the photobleaching process would be helpful.

The relationship between light dose and bleaching is exponential, f(r) = -1 +exp(- *k* I(r) T). The term ‘I(r) * T’ is the light dose (I, intensity; T, time). The term *k* was fit based on empirical data (Figure 3C). Fits were done separately for GFP (473 nm light) and mCherry (594 nm light). We have added this information to the Materials and methods (subsection “Photostimulation”). *k* is dependent on the bleaching cross section of the fluorescent molecule.

3) If the radius of laser light scatter is ~200 microns, then it is still confusing why the viral injection, which has a radius of ~250 microns (i.e. greater than the light) should restrict the spread of inhibition compared to in the transgenic condition. This result suggests that the authors are underestimating the spread of their light.

See response to point 1 above. The light spreads beyond 200 microns, but the intensity is relatively low. Moreover, dendrites and axons of GABAergic neurons are photosensitive. Quantitatively explaining the spatial spread of optogenetic silencing will require modeling that considers the spatial distribution of light in tissue, neuronal morphology, and ChR2 expression levels, activation thresholds, and circuit connectivity.

4) The terminology regarding light levels in the revised manuscript and rebuttal letter is quite inconsistent, with moderate meaning different things (1 mW or 20 mW) in different contexts. This terminology should be standardized throughout the paper, especially in relation to light levels required for biological effects.

We have revised the text so that the terminology is consistent. We now only use ‘moderate’ for power level 1.5 mW. 20 mW was in reference to light levels used in the photobleaching experiments. To avoid confusion, we now use the terminology ‘light dose’ in ‘mW x min’ to reflect the total amount of light delivered in photobleaching.

5) It would be valuable to add some discussion regarding how these optogenetic silencing approaches seem to be extremely sensitive such that there is not much dynamic range for the manipulation, regarding e.g. the possibility to explore the behavioral consequences of different degrees of cortical suppression. On a related note, the new limited data provided in the rebuttal on silencing with low level light delivery should be included in the revised paper.

We have included the low power level data from PV-cre x ReaChR mice in the revised manuscript. In addition, we now also include silencing data at low light doses for VGAT-ChR2-EYFP mice (Figure 1E). We have added a discussion on the sensitivity of optogenetic experiments (Discussion, seventh paragraph).

ChR-assisted photoinhibition and Cl^-^ channel mediated directed photoinhibition are highly sensitive, producing similarly complete silencing near the laser center across a large range of laser powers (photoinhibition, 1.5 – 10 mW, Figure 1E). However, we note that the spatial spread of photoinhibition differed substantially across laser powers (Figure 5 and 6). Higher laser power removed spikes over a larger cortical region. This power-dependent spatial spread produces graded loss-of-function manipulations in vivo. For example, photoinhibiting the barrel cortex induced increasingly larger behavioral effects with increasing laser power across a large power range (1.5-10 mW) (Guo et al., 2014b).